# DNA Ligase C and Prim-PolC participate in base excision repair in mycobacteria

Przemysław Płociński[1,2], Nigel C. Brissett[1], Julie Bianchi[1,4], Anna Brzostek[2], Małgorzata Korycka-Machała[2], Andrzej Dziembowski[3], Jarosław Dziadek[2] & Aidan J. Doherty[1]

Prokaryotic Ligase D is a conserved DNA repair apparatus processing DNA double-strand breaks in stationary phase. An orthologous Ligase C (LigC) complex also co-exists in many bacterial species but its function is unknown. Here we show that the LigC complex interacts with core BER enzymes in vivo and demonstrate that together these factors constitute an excision repair apparatus capable of repairing damaged bases and abasic sites. The polymerase component, which contains a conserved C-terminal structural loop, preferentially binds to and fills-in short gapped DNA intermediates with RNA and LigC ligates the resulting nicks to complete repair. Components of the LigC complex, like LigD, are expressed upon entry into stationary phase and cells lacking either of these pathways exhibit increased sensitivity to oxidising genotoxins. Together, these findings establish that the LigC complex is directly involved in an excision repair pathway(s) that repairs DNA damage with ribonucleotides during stationary phase.

[1] Genome Damage and Stability Centre, School of Life Sciences, University of Sussex, Brighton, BN1 9RQ, UK. [2] Institute of Medical Biology, Polish Academy of Sciences, Lodowa 106, 93-232 Lodz, Poland. [3] Institute of Biochemistry and Biophysics, Polish Academy of Sciences, Pawińskiego 5A, 02-106 Warsaw, Poland. [4] Present address: Department of Oncology-Pathology, Cancer Center Karolinska, Karolinska Institutet, R8:04, Karolinska Universitetssjukhuset Solna, 171 76 Stockholm, Sweden. Przemysław Płociński and Nigel C. Brissett contributed equally to this work. Correspondence and requests for materials should be addressed to A.J.D. (email: ajd21@sussex.ac.uk)

In bacteria, canonical primer synthesis during DNA replication is carried out by enzymes from the DnaG superfamily[1, 2]. In contrast, priming of replication in archaea and eukaryotes is performed by members of the archaeo-eukaryotic primase (AEP) superfamily[3, 4]. However, AEPs are also widely distributed in most bacterial species[4], where they have evolved to fulfil divergent roles and have recently been reclassified as a family of polymerases called primase-polymerases (Prim-Pols) to better reflect their evolutionary origins and more diverse roles in DNA metabolism[4]. The best characterised bacterial AEP is Prim-PolD (PolDom) that forms part of a multifunctional non-homologous end-joining (NHEJ) DNA break repair complex called Ligase D (LigD). In mycobacterial LigD, an AEP is fused to phosphoesterase and ATP-dependent DNA ligase domains that, together with the Ku repair factor, coordinate the sequential synapsis, processing and repair of double-strand breaks (DSBs) in stationary phase[5–9]. However, in many other species these domains are encoded by separate operonically associated genes[6, 10].

Many bacterial species, including *Actinobacteria*, encode multiple ATP-dependent DNA ligases and Prim-Pols similar to those found in mycobacterial LigD. Exemplary classes of Prim-Pols and DNA ligases in selected gram-positive bacteria are shown in Supplementary Fig. 1. *Mycobacterium smegmatis* encodes four distinct primase-polymerases. Although it is known that the Prim-PolD subunit of LigD is involved in the NHEJ repair complex, the roles of the other stand-alone Prim-Pols remain unknown. One orthologue, MSMEG_6301 (Prim-PolC/LigC Pol/PolD) is encoded in the genomic proximity of two DNA ligase genes (LigC1: *msmeg_6302* and LigC2: *msmeg_6304*). Although LigD's role in NHEJ-mediated repair is firmly established[7, 9, 11], the pathways in which these other ligases and Prim-Pols operate in remain unclear. Due to their similarities with the NHEJ complex, it was proposed that DNA ligase C and operonically associated Prim-PolC (LigC-Pol or PolD1) function as a "back-up" complex for the LigD pathway[12, 13, 14], although this has not been proven. While the basic biochemical characteristics of Prim-PolC and PolD2, another closely related AEP, were partially described in a previous study[12], Prim-PolC failed to act as a "back-up" of Prim-PolD during DSB repair. Another study reported that LigC was responsible for ligation of ~20% of DSBs

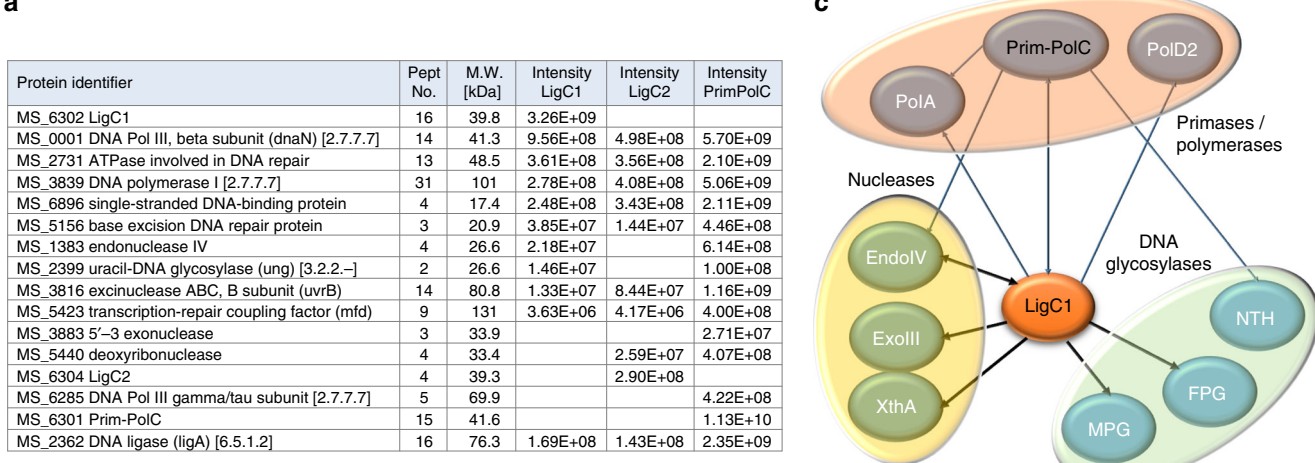

| Protein identifier | Pept No. | M.W. [kDa] | Intensity LigC1 | Intensity LigC2 | Intensity PrimPolC |
|---|---|---|---|---|---|
| MS_6302 LigC1 | 16 | 39.8 | 3.26E+09 | | |
| MS_0001 DNA Pol III, beta subunit (dnaN) [2.7.7.7] | 14 | 41.3 | 9.56E+08 | 4.98E+08 | 5.70E+09 |
| MS_2731 ATPase involved in DNA repair | 13 | 48.5 | 3.61E+08 | 3.56E+08 | 2.10E+09 |
| MS_3839 DNA polymerase I [2.7.7.7] | 31 | 101 | 2.78E+08 | 4.08E+08 | 5.06E+09 |
| MS_6896 single-stranded DNA-binding protein | 4 | 17.4 | 2.48E+08 | 3.43E+08 | 2.11E+09 |
| MS_5156 base excision DNA repair protein | 3 | 20.9 | 3.85E+07 | 1.44E+07 | 4.46E+08 |
| MS_1383 endonuclease IV | 4 | 26.6 | 2.18E+07 | | 6.14E+08 |
| MS_2399 uracil-DNA glycosylase (ung) [3.2.2.–] | 2 | 26.6 | 1.46E+07 | | 1.00E+08 |
| MS_3816 excinuclease ABC, B subunit (uvrB) | 14 | 80.8 | 1.33E+07 | 8.44E+07 | 1.16E+09 |
| MS_5423 transcription-repair coupling factor (mfd) | 9 | 131 | 3.63E+06 | 4.17E+06 | 4.00E+08 |
| MS_3883 5'–3 exonuclease | 3 | 33.9 | | | 2.71E+07 |
| MS_5440 deoxyribonuclease | 4 | 33.4 | | 2.59E+07 | 4.07E+08 |
| MS_6304 LigC2 | 4 | 39.3 | | 2.90E+08 | |
| MS_6285 DNA Pol III gamma/tau subunit [2.7.7.7] | 5 | 69.9 | | | 4.22E+08 |
| MS_6301 Prim-PolC | 15 | 41.6 | | | 1.13E+10 |
| MS_2362 DNA ligase (ligA) [6.5.1.2] | 16 | 76.3 | 1.69E+08 | 1.43E+08 | 2.35E+09 |

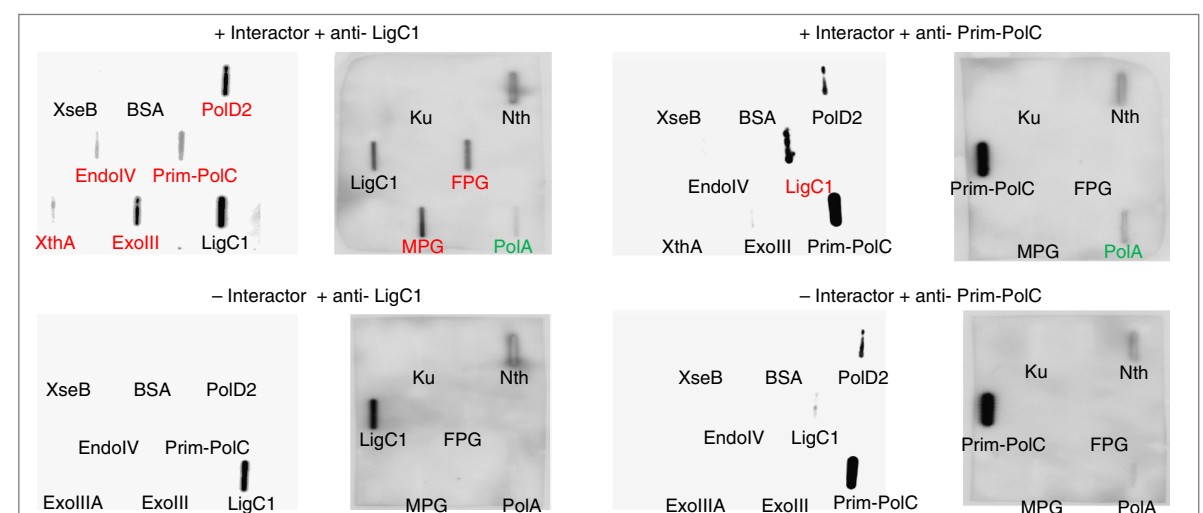

**Fig. 1** Interactions between Prim-PolC, LigC proteins and base excision repair elements. **a** A table showing the DNA repair-associated preys that co-purified in an eGFP-facilitated affinity purification experiment using LigC and Prim-PolC as baits. **b** Base excision repair enzymes were purified as recombinant proteins and interactions with Prim-PolC and LigC1 were confirmed by slot-blot analysis, where positive interactions are marked with red, possible weak associations with green and negative interactions with black font, respectively. **c** Verified interactions are summarised in a schematic diagram showing that LigC is the major scaffolding protein involved in multiple protein complex formation

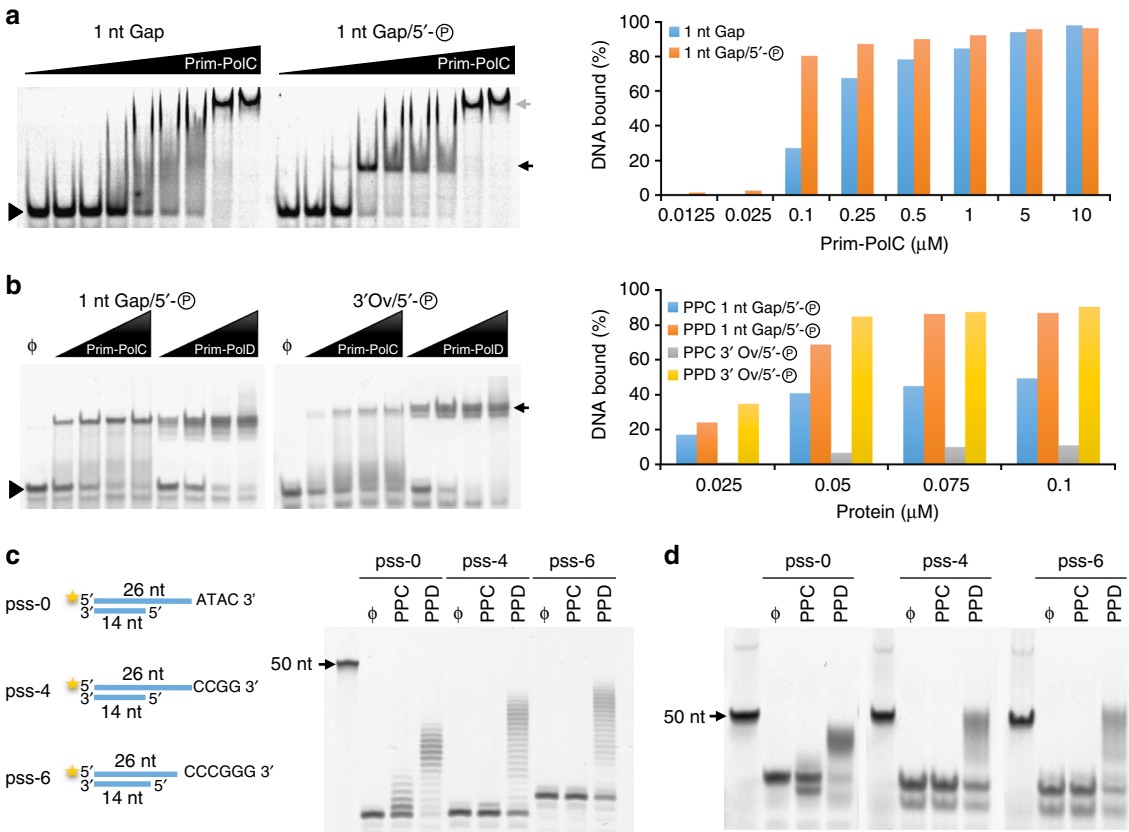

**Fig. 2** DNA binding activity of Prim-PolC. **a** Prim-PolC prefers to bind to 5′ phosphorylated gaps. In EMSAs, mixtures of a single nucleotide gap containing substrate (30 nM), with or without phosphorylation of the 5′ end of the lesion, and 0–10 μM Prim-PolC were incubated for 20 min and resolved on a native polyacrylamide gel. A filled triangle indicates unshifted probe, whereas the black arrow probably indicates a binary complex of Prim-PolC and DNA. The grey arrow indicates multiple protein monomers bound to the probe. Quantification of the EMSA data is presented. For each Prim-PolC concentration the percentage of DNA bound (in relation to the total DNA) was calculated and compared for EMSAs containing Prim-PolC binding a 1 nucleotide gap with or without 5′-phosphorylation. **b** Prim-PolC prefers to bind to gaps rather than overhangs unlike Prim-PolD. This EMSA uses 30 nM of 5′-fluorescein labelled 36-mer containing a 5′ phosphorylated single nucleotide gap or 30 nM of 5′-fluorescein labelled 36-mer containing a single-stranded 3′ overhang given by phosphorylation of the 5′-end of the gap and no primer strand. The triangles and arrow are as in **a**. Using the same conditions as **a**, this time comparing Prim-PolC to *Mtu* Prim-PolD (25, 50, 75 and 100 nM). Quantification of the EMSA data are presented. For each enzyme concentration the percentage of DNA bound (in relation to the total DNA) was calculated and compared for EMSAs containing Prim-PolC or Prim-PolD binding a 5′-phosphorylated 1 nucleotide gap or a 3′-overhang with a 5′-phosphorylated downstream strand. **c** Prim-PolC is not an NHEJ polymerase. A schematic of 5′-fluorescein labelled substrates used in a MMEJ activity assay. The pss-number refers to the number of bases at the end of the 3′overhang that can base pair with itself. A primer extension assay, 30 nM of the denoted substrate extended by 300 nM enzyme (PPC-Prim-PolC, PPD-*Mtu* Prim-PolD) in the presence of 250 μM dNTPs mix for 30 min at 37 °C. The products are resolved on a 15% denaturing gel. **d** The same reaction products from **c**, after treatment with Proteinase K and run on a 12% non-denaturing gel

in a LigD mutant strain containing a mutation disrupting the ligase function but possessing functional primase and exonuclease domains that can still participate in the NHEJ[14].

Here we sought to elucidate the role(s) of LigC, and its associated polymerase (Prim-PolC), in mycobacterial DNA metabolism. We show that components of the LigC complex act together to preferentially fill in short DNA gaps with ribonucleotides (rNTPs), followed by sealing of the resulting nicks. We identify that LigC operon-encoded proteins associate with components of the base excision repair (BER) apparatus in vivo and, together with the excision repair machinery, form repair complexes that remove damaged DNA and abasic sites and fill in resulting gaps with rNTPs. Structural studies reveal that a conserved C-terminal element of Prim-PolC forms an extended surface surrounding the active site, which likely supports gap recognition and synthesis. Components of the LigC complex are expressed upon entry into stationary phase, when the intracellular pool of dNTPs is naturally depleted, necessitating the preferential incorporation of

rNTPs after lesion removal. Abrogation of the LigC complex, results in an increased sensitivity of mycobacterial cells to oxidative damaging agents. LigD-deficient cells are also sensitive to oxidative damaging agents, uncovering an unexpected duel role in both excision and DSB repair. Together, these findings establish that the LigC complex operates in excision repair pathways that specifically mends lesions and single-strand breaks in stationary phase.

## Results

**DNA Ligase C complex interacts with BER enzymes in vivo.** Although LigD-associated Prim-Pols and ligases facilitate NHEJ repair of DSBs in bacteria, the functions of closely related orthologues remain unclear. To investigate the cellular roles of Prim-PolC, and its operonically associated DNA ligases (LigC1 and LigC2), we first sought to define their physical associations with other proteins/complexes in mycobacterial cells. We

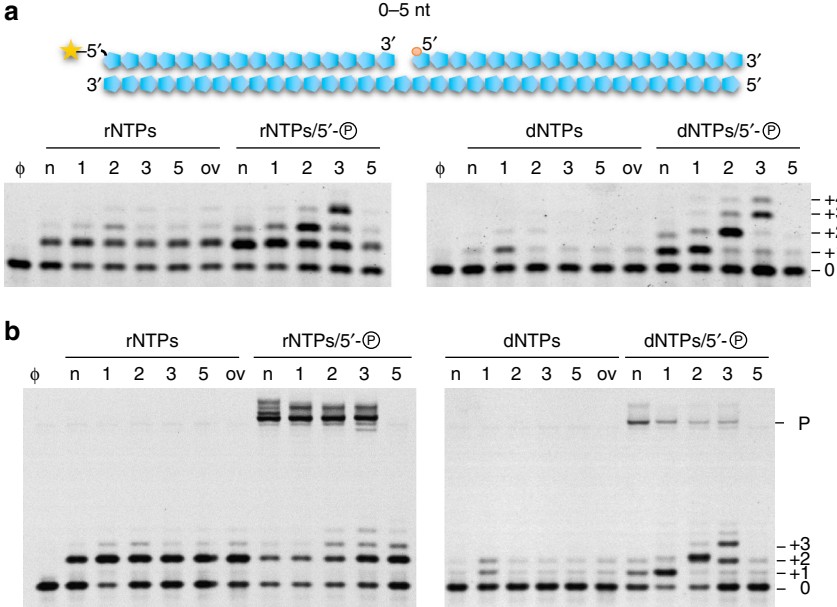

**Fig. 3** Short gap-filling activity of Prim-PolC in complex with LigC1. **a** Prim-PolC prefers to insert rNTPs into short gaps. In primer extension assays, 30 nM of 5′-fluorescein labelled 36-mer containing a nick (n), a single-stranded gap of various length (indicated as numbers), with or without phosphorylation of the 5′-end or lacking the 5′-end strand thus forming an overhang (ov), was extended by Prim-PolC (300 nM) in the presence of a 250 μM dNTPs or rNTPs mix for 5 min at 37 °C. **b** The preferred products from Prim-PolC gap-filling are ligatable. For gap filling and ligation, LigC1 (300 nM) was added to the primer extension reactions (same as **a**) and were incubated for an extra 45 min at 37 °C to produce a repair product (P)

employed an eGFP-tag (enhanced green fluorescent protein) facilitated co-purification approach, in which each of the proteins (Prim-PolC, LigC1 and LigC2) were fused to eGFP to serve as baits for the purification and identification of cellular partners. Purified complexes were cross-linked with a reversible cross-linker, 3,3′-dithiobis (sulfosuccinimidyl propionate) (DSSP), washed and subjected to mass spectrometry (MS) analysis[15]. Complexes were purified from cells grown to late stationary phase. As expected, tagged LigC and Prim-PolC were among most abundant "hits" in the MS samples, supporting the specificity of the purification process. A list of polypeptides that co-purified with each bait, known to be associated with DNA repair processes, are shown in Fig. 1a. A full list of all co-purified proteins is provided in Supplementary Data 1. Significantly, all three bait proteins co-purified with multiple repair enzymes, including the major DNA glycosylases and nucleases involved in base excision repair (BER) repair pathways. Reverse experiments using mycobacterial eGFP-tagged endonuclease IV as bait in *M. bovis* BCG resulted in co-purification of LigC, confirming that these two proteins form a complex in vivo (Supplementary Data 2). Together, these in vivo studies reveal that Prim-PolC and LigC interact with BER components, suggesting that they function primarily in the repair of damaged bases, abasic sites and single-strand breaks (SSBs).

**DNA ligase C complex interacts with BER enzymes in vitro.** To validate the interactions of LigC1 and Prim-PolC with components of the BER machinery identified in our pull-down studies, we expressed and purified recombinant forms of each of these BER enzymes (Supplementary Fig. 2). Taking advantage of Prim-PolC and LigC1-specific antibodies, a slot blotting-based methodology was employed to authenticate the interactions between the identified proteins. We also used this approach to address if Prim-PolC or LigC1 interacted with Ku, to determine if they also function in NHEJ repair. Mono-functional DNA glycosylase (MPG) was purified alongside bifunctional glycosylases (FPG and Nth), that possess abasic site lyase activity, as well as several of the

key end-processing nucleases including: exodeoxyribonuclease VIIB (ExoVIIB), endonuclease IV (EndoIV) and both exonuclease III paralogues (ExoIII, XthA) from *M. smegmatis*. PolD2, the Prim-Pol with closest similarity to Prim-PolC, as well as the major BER repair polymerase PolA, were also purified for these in vitro interaction studies. Recombinant proteins were spotted onto nitrocellulose membrane and incubated with either Prim-PolC or LigC1, respectively, followed by extensive washing and subsequent detection by western blotting using Prim-PolC/LigC1-specific antibodies. We observed significant signals with many of the BER proteins incubated with LigC1 (Fig. 1b). MPG and FPG (MutM) glycosylases exhibited the strongest interactions with LigC1. LigC1 also interacted with EndoIV, ExoIII and XthA end-processing enzymes and both Prim-PolC and PolD2. A summary of the major interactions identified in these studies is shown in Fig. 1c. In contrast, Prim-PolC only interacts with LigC1 but not with either Ku or ExoVIIB, which therefore served as control proteins for these experiments. Together, these data establish that LigC appears to act as a key scaffolding protein, akin to ligase III in eukaryotes[16, 17], required for sequential recruitment of BER repair enzymes and therefore potentially coordinating the processing and repair of lesions and SSBs in vivo[18–20]. To further validate the interactions with the glycosylase, we performed pull-down assays of LigC1 with the two potentially interacting glycosylases, FPG and MPG and observed a strong association of tag-free LigC1 with FPG coated His-trap beads and a weaker interaction with MPG (Supplementary Fig. 3).

Previously, comparative sequence database mining analysis of bacterial operons uncovered a genetic link between LigD and Ku, which led to the identification of the NHEJ pathway in prokaryotes[21, 22]. Similar in silico comparative operon analysis of many mycobacterial genomes revealed that the LigC and Prim-PolC genes are co-transcribed with the major base excision repair bifunctional glycosylase, FPG, in operons present in *M. avium*, *M. intracellulare* and *Mycobacterium JS623*, a close relative of *M. smegmatis*. Notably, these operons are also located within close genomic proximity of other DNA excision repair enzymes (PolA

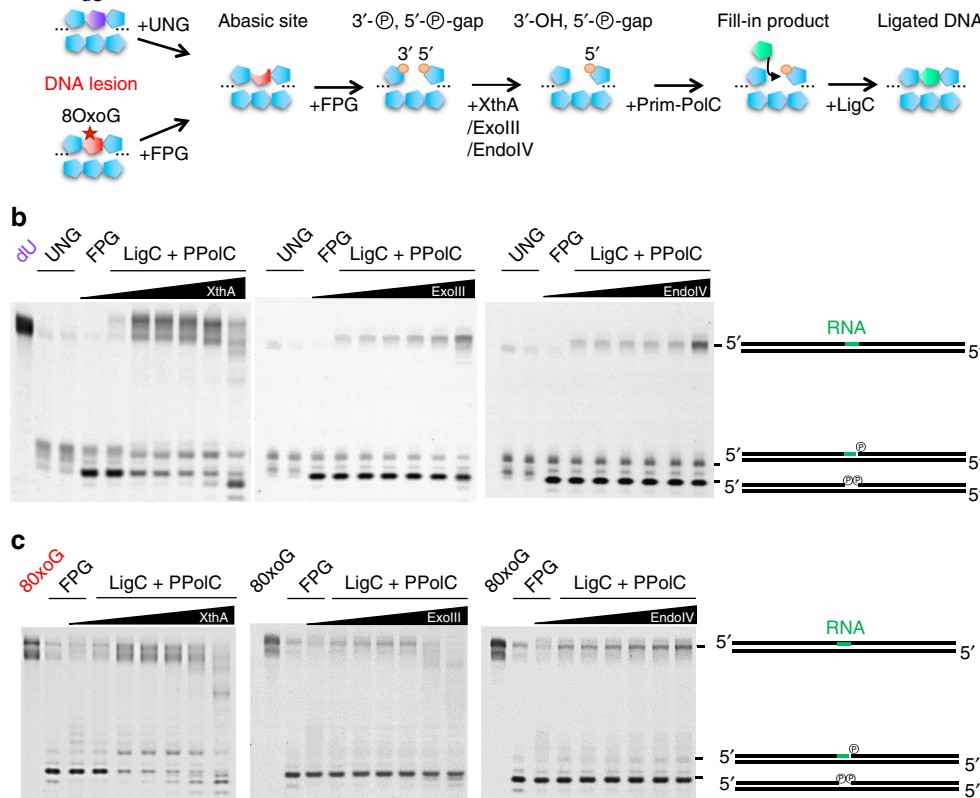

**Fig. 4** Reconstitution of a Prim-PolC - LigC1-dependent BER repair complex in vitro. **a** Schematic representation of BER repair reactions showing repair intermediates generated by subsequent DNA repair activities of glycosylases, end-processing enzymes, gap-filling and ligation. **b, c** In vitro repair of deoxyuracil- **b** or 8-oxoguanine- **c** containing DNA substrates. 30 nM of 5′-fluorescein labelled ds 36-mer containing a central lesion was processed by uracil glycosylase or directly by recombinant mycobacterial FPG (for 8-oxoguanine), to produce a resulting abasic site intermediate processed into a 3′,5′-biphosphorylated gap by the abasic site lyase activity of FPG. Reactions were carried out with addition of a 250 μM dNTPs or rNTPs mix for 30 min at 37 °C. Gap ends were next processed, including removal of the 3′ phosphate, by addition of various amounts of either XthA, ExoIII or EndoIV for 5 min at 37 °C. Subsequently, a mixture of Prim-PolC and LigC1, 300 nM of each, was added to provide gap filling and ligation activities for 45 min at 37 °C

and UvrB; Supplementary Fig. 3). Together, these findings further strengthen a functional connection between LigC-related proteins and BER processes in mycobacteria.

**Prim-PolC preferentially binds to short gapped DNA.** Following validation of the association of Prim-PolC and LigC1 with components of the BER machinery, we proceeded to biochemically characterize the capacity of these enzymes to process DNA repair intermediates that arise during the processing of BER substrates, e.g., abasic sites. We first performed electrophoretic mobility shift assays (EMSAs) to determine the optimal DNA substrates for binding and processing by Prim-PolC. The most common BER intermediates are short DNA gaps containing a phosphate moiety on the 5′ terminus, that arise from base excision and subsequent abasic site removal. We therefore tested Prim-PolC's capacity to bind to an array of phosphorylated and non-phosphorylated DNA substrates, containing various gap lengths. Notably, Prim-PolC's binding affinity was significantly stimulated by the presence of a 5′-phosphate group within a gap and it bound most avidly to short gaps of 1–3 nucleotides (Fig. 2a; Supplementary Fig. 4a). In contrast with its NHEJ-specific orthologue, Prim-PolC bound much more weakly to DNA substrates containing overhangs (Fig. 2b).

NHEJ Prim-Pols bind strongly to 5′ phosphorylated DNA overhangs[23] and can directly induce break synapsis[23–26]. To determine if Prim-PolC also possesses end-synapsis activity, we employed a microhomology-mediated end-joining (MMEJ) assay that utilises substrates that mimic DSBs to measure this activity in vitro[27]. As expected, Prim-PolD efficiently promoted break synapsis and subsequent in-trans extension of MMEJ intermediates (Fig. 2c, d). In contrast, Prim-PolC did not exhibit significant MMEJ activity, indicating that DSBs are not preferred substrates for this enzyme. This identifies another significant difference in substrate specificities between these two classes of Prim-Pols.

To cross-validate the observed DNA substrate binding affinities, we carried out additional DNA binding studies using polyA-tailed DNA substrates coated onto magnetic particles. Beads pre-coated with different DNA substrates were incubated with whole-cell mycobacterial lysates, washed extensively and bound preys were identified by mass spectrometry. As expected, LigD and Ku were both found to co-purify on DSB-like substrates with free ends. However, neither Ku or LigD bound to this substrate when the terminal nucleotides were blocked with S-bond. Notably, Prim-PolC and LigD, but not Ku, co-purified on a substrate containing a single nucleotide gap, suggesting that both may contribute to the repair of such intermediates in vivo (Supplementary Fig. 4b, Supplementary Data 3).

**LigC complex repairs short gapped DNA intermediates.** To establish the functional importance of Prim-PolC's preferential gap binding activity, we next performed primer-template extension assays using gapped substrates (1, 2, 3 or 5 nt gap) in the

**Table 1 Data collection and refinement statistics (molecular replacement)**

|  | *Msm* Prim-PolC |
|---|---|
| *Data collection* |  |
| Space group | $P_1$ |
| Cell dimensions |  |
| $\;\;a, b, c$ (Å) | 51.55, 56.43, 64.45 |
| $\;\;\alpha, \beta, \gamma$ (°) | 97.17, 100.2, 90.64 |
| Resolution (Å) | 44.77 (1.84)[a] |
| $R_{sym}$ or $R_{merge}$ | 0.09 (0.60) |
| $I/\sigma I$ | 6.8 (1.7) |
| Completeness (%) | 96.9 (95.4) |
| Redundancy | 2.6 (2.6) |
| *Refinement* |  |
| Resolution (Å) | 44.77 (1.84) |
| No. of reflections | 59,594 (5894) |
| $R_{work}/R_{free}$ | 0.1858/0.2092 |
| No. of atoms | 5824 |
| $\;\;$Protein | 5254 |
| $\;\;$Water | 570 |
| *B-factors* |  |
| $\;\;$Protein | 28.81 |
| $\;\;$Water | 37.11 |
| R.M.S. deviations |  |
| $\;\;$Bond lengths (Å) | 0.005 |
| $\;\;$Bond angles (°) | 0.743 |

[a]From 1 crystal. Values in parentheses are for highest resolution shell

presence of intracellular levels of divalent cations. Consistent with the DNA binding results, Prim-PolC most efficiently processed short DNA gaps (Fig. 3a), with its polymerisation activity significantly stimulated by the presence of a phosphate group at the 5′ terminus of the gap (Fig. 3a). In common with Prim-PolD, Prim-PolC also preferentially inserted rNTPs during gap-filling synthesis (Fig. 3a). This intrinsic preference for NTPs over dNTPs was quantified using a nucleotide competition assay (Supplementary Fig. 5a)[23]. This assay revealed that its nucleotide preference factor (F) was ~190-fold in favour of rNTP incorporation, which compares to an F-value of ~70-fold for *Mt* Prim-PolD (PolDom)[23].

The implication of this finding is that gap-filling synthesis by Prim-PolC would preferentially place RNA, or at least a single rNTP, on the 3′ side of the nick that must then be ligated to DNA on the opposite 5′ side. To test this prediction, we evaluated the ligation specificity of LigC1 using nicked DNA substrates containing either DNA or RNA on the 3′ side and a phosphorylated DNA strand on the 5′ side. LigC1 preferentially ligated hybrids of RNA and DNA within the repaired gap, rather than more typical DNA:DNA nicks (Supplementary Fig. 5b), consistent with the previously reported ligation specificity of LigD[10, 14]. As repair synthesis by Prim-PolC would result in the production of such hybrid RNA:DNA nicks in vitro, we next tested whether the two enzymes co-operatively repair short DNA gaps. Together, Prim-PolC and LigC1 efficiently filled in and ligated short (1–3 nt) gaps with RNA but were much less efficient at repairing longer DNA gaps (Fig. 3b), consistent with the reduced DNA binding and extension activities of Prim-PolC observed previously on these substrates (Fig. 3a and Supplementary Fig. 4a).

**Reconstituting a LigC-dependent BER repair pathway in vitro.** To establish whether the LigC apparatus operates as part of a functional BER complex, we sought to reconstitute a complete excision repair pathway in vitro using Prim-PolC, LigC1 and the interacting BER proteins. We employed oligonucleotide substrates containing either deoxyuracil (dU) or 8-oxoGuanine (8-oxoG) incorporated in a central position of the labelled DNA strand to enable us to follow the consecutive enzymatic processing steps of BER (Fig. 4a). dU-containing substrates were initially treated with UDG glycosylase and the resulting abasic site substrate was then tested with the same enzyme combinations also assayed on 8-oxoG containing substrates. We observed that FPG removed the oxidatively damaged DNA base and processed the resulting abasic site with its lyase activity (Fig. 4b, c). The 3′ end of the gap was subsequently processed by either one of the 3′-phosphatase and/or 3′-5′ exonuclease activities of EndoIV, ExoIII or XthA. The resulting DNA gap was subsequently filled in with RNA by Prim-PolC and, finally, the resulting nick was ligated by LigC1. We observed that EndoIV was the most beneficial nuclease for ensuring proper repair in vitro as it lacked strong 3′-5′ exonuclease activity. Together, these findings establish that BER end-processing enzymes, shown to interact with LigC1 in vivo, facilitate LigC complex-mediated repair of lesions arising from abasic site formation in vitro.

**Structure of a mycobacterial LigC-associated Prim-Pol.** Although Prim-PolC shares many common features with its NHEJ orthologue, such as 5′ phosphate recognition and gap-filling synthesis, this enzyme is likely to have evolved distinct structural features to facilitate its bespoke role in excision repair. To define molecular differences between these distinct Prim-Pols, we elucidated the crystal structure of *Ms* Prim-PolC (Table 1) and compared it to a NHEJ-specific Prim-Pol. Prim-PolC retains the same overall fold of the mycobacterial Prim-PolD (PolDom) structure[23–26], which can be superposed with an RMSD of 1.712 Å (over 270 aligned positions, Fig. 5a; Supplementary Fig. 6a). Shared structural features include the phosphate recognition pocket containing the conserved basic residues (K23, K35 and N20) that facilitate this key DNA interaction. Prim-PolC also retains Loops 1, which is critically required to orient the template/ 3′ overhanging strand, and Loop 2 that accepts and positions the incoming primer strand from the adjacent side of the break and also facilitates the binding/activation of the second metal ion in the active site[23–26]. These loops are critical for Prim-PolD's engagement with the break termini, promoting end-synapsis/ MMEJ and are required for activation of the catalytic centre to allow DNA synthesis to commence[25, 26].

Despite these similarities, Prim-PolC orthologues possess a strictly conserved C-terminal extension whose function has not been ascribed (Fig. 5b). In the Prim-PolC structure, we observe that this additional C-terminal region (aa 294–336) consists of a conserved alpha helix (residues G299–R311), which lies across alpha helix 4 after which the peptide chain becomes an unstructured loop (Fig. 5a, b). Notably, this structural element, termed Loop 3 (residues G313–P333), is positioned between the two prominent surface loops (1 and 2) that together form a continuous surface on one side of the active site pocket (Fig. 5c, Supplementary Fig. 6b). Apical Loop 1 residues, normally involved in template strand orientation[23–26], are not conserved and are instead repurposed for locking Loop 3 into position (Supplementary Figs. 6b, 7). Residue P94 packs against a pocket formed of residues Y321, P322, K323 and M324 from Loop 3. This also results in Loop 1 adopting a position more proximal to the active site. Loop 2 residue W219 interacts with Loop 3 residues P316, Y317, P333 and K335 via a mixture of hydrophobic and polar interactions (Supplementary Figs. 6b, 7). Again, this has the effect of locking Loop 3 into its current orientation, and also reorients Loop 2 into a more distal position

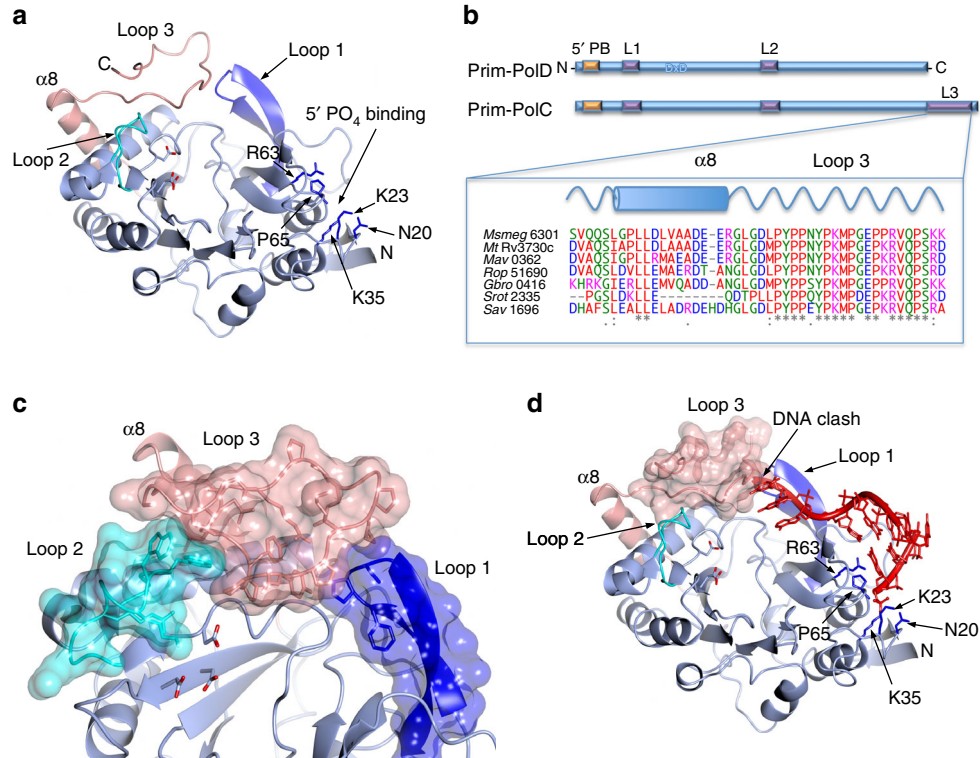

**Fig. 5** Crystal structure of Prim-PolC. **a** A ribbon diagram representation of the crystal structure of Prim-PolC (PDBID: 5OP0). The previously characterised NHEJ Prim-Pol core is coloured sky blue (Fig. S6a), with the conserved loop structures from this family coloured dark blue and cyan for Loops 1 and 2, respectively. The catalytic aspartate side-chains are represented with the carboxylic oxygens shown in red. Side-chains involved in forming the 5′ phosphate binding pocket are depicted in dark blue. The Prim-PolC specific C-terminal extension is shown in pink, comprising α8 and Loop 3. **b** Schematic representation comparing the primary structures of Prim-PolD and Prim-PolC. Conservation of the C-terminal extension of Prim-PolC among selected organisms are highlighted by a sequence alignment of this region. **c** Ribbon diagram representation of the Prim-PolC crystal structure highlighting the molecular contacts formed between the conserved loops (1 and 2) and the additional Loop 3. The loops are further represented as translucent solvent accessible surfaces colour matched to the specific loop element. **d** Superposition of DNA from the NHEJ Prim-PolD MMEJ complex (PDBID: 4MKY) onto the Prim-PolC structure. The path of the templating DNA from the *in-trans* structure clashes with Loop 3

to the active centre, compared to the existing Prim-PolD MMEJ structure[25]. A consequence of Loop 2's interaction with Loop 3 is the fixing of Loop 2 into its current orientation. The effects of limiting the flexibility of Loop 2, and therefore the conserved arginine 224 (equivalent to R220 in *Mt* Prim-PolD), may have a significant effect on Prim-PolC's catalytic activity.

Superposition of Prim-PolC onto the structure of a DNA bound Prim-PolD MMEJ complex revealed that the 5′ phosphate fits into the conserved binding pocket and the dsDNA is also well positioned into the structure (Fig. 5d and Supplementary Fig. 6c, d). The first three bases of the single-stranded template strand/3′ overhang are within contacting distance of Loop 1. However, the 3′ terminal bases of this strand clash with Loop 3 in the Prim-PolC structure (Fig. 5d, Supplementary Fig. 6c). This clash suggests that a possible key role of the insertion is to accommodate a continuous template strand that is reoriented towards the active site thus positioning gapped substrates, as opposed to NHEJ intermediates, in a more optimal position. This repositioning ensures, together with Loops 1 and 2, that the primer strand is correctly oriented into the active site in preparation for gap-filling synthesis although this prediction remains to be proven.

**LigC or LigD mutants are sensitive to oxidative DNA damage.** Collectively, our interaction and in vitro reconstitution studies strongly implicate Prim-PolC and LigC, and their associated

factors, in the repair of lesions and SSBs processed via abasic site intermediates. As such lesions are formed in large quantities during all stages of the cell cycle, we next sought to determine if this repair pathway functions throughout the cell cycle or, like LigD-dependent repair, is specific to a particular stage. To address this, we examined the expression profiles of Prim-PolC and LigC1 at different phases of the growth cycle (early, mid, stationary) using specific antibodies. This revealed that the levels of both proteins peaks during early stationary phase (Fig. 6a), indicating that this pathway preferentially operates as cells begin to enter stationary phase.

To evaluate their roles in excision repair processes in vivo, we deleted *prim-polC* (Δ*prim-polC*) and *ligC1, ligC2* (Δ*ligC1, C2*) in *M. smegmatis*. To test whether deletion of *ligD* and *prim-polC* genes have a cumulative effect or the proteins can fully compensate for one another's activities, we generated single (Δ*LigD*) and double (Δ*ligD, prim-polC*) deletion mutants. We also created a *polA* mutant complemented with truncated PolA, lacking functional DNA polymerase domain but maintaining its essential 3′ exo activity[28], (Δ*polA*:: PolA1-755 aa) along with *prim-polC* and *ligD* mutants in a *polA* deficient background (Δ*prim-polC, polA*:: PolA1-755 aa; Δ*prim-polC, ligD, polA*:: PolA1-755 aa). Cells were grown to stationary phase before exposing them to genotoxic agents, to assess the oxidative damage sensitivity of each strain. We tested several oxidising agents and observed decreased viabilities for Δ*prim-polC* and Δ*ligC1,C2* strains exposed to oxidising agents (Fig. 6b, Supplementary

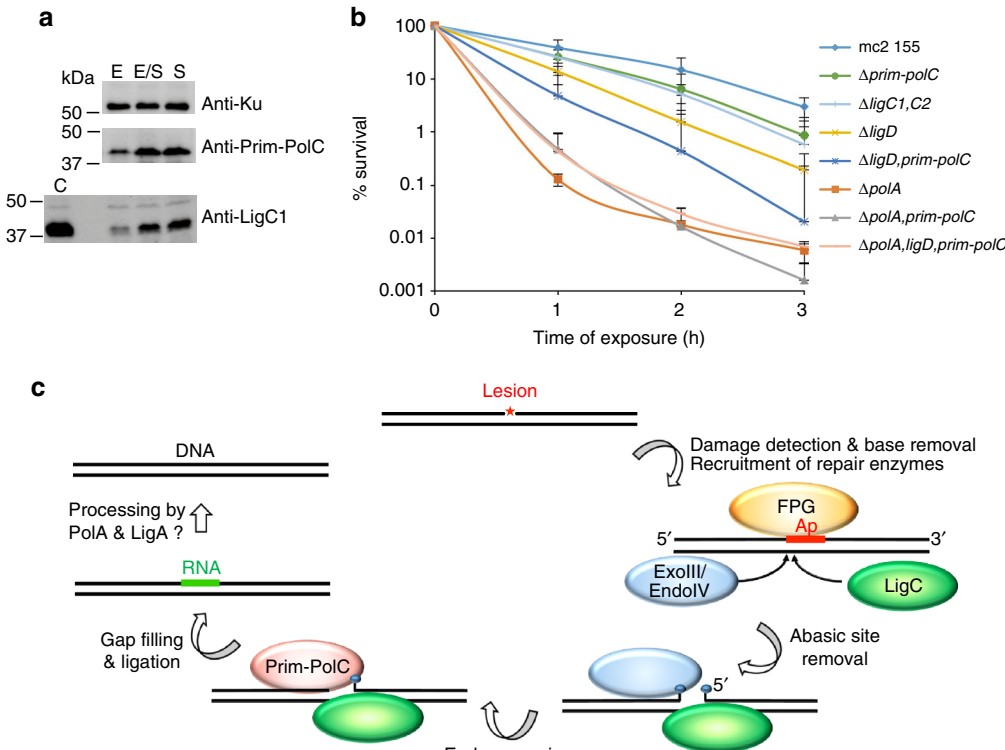

**Fig. 6** Expression of Prim-PolC and LigC1 in vivo and phenotypes associated with their deletion. **a** Expression profiling using western blotting. Specific antibodies raised against LigC1, Prim-PolC and Ku proteins were used to probe for corresponding proteins in whole-cell lysates obtained from *M. smegmatis* cells grown to various growth phases including, exponential (E"), early stationary (E/S) and stationary (S) phase. **b** A plot of colony forming units (CFU) data showing the survival of wild-type (MC 155) or mutant cells, lacking Prim-Pols and/or ligases, treated with 10 mM cumene hydroperoxide. Cells were plated at 1, 2 or 3 hours after treatment and the percentage survival is calculated against untreated control cells, treated with diluted buffer instead of peroxide. Data are representative of the mean of five individual experiments and error bars show the standard deviation. **c** A model of the repair of a damaged base by the LigC-dependent excision repair complex. The damage is initially recognised by specific mono or bifunctional glycosylases, whose activity results in the formation of an abasic site intermediate. LigC is recruited early to the lesion to coordinate later repair steps and mediate multi-protein repair complex formation. Short gaps produced by processing of the lesion by the BER machinery are then filled in with RNA by Prim-PolC and the resulting nick is ligated by LigC. RNA is probably later excised and replaced with DNA catalysed by PolA and the nicked DNA intermediate sealed by LigA

Fig. 8), particularly cumene hydroperoxide and tert-butyl hydroperoxide. We observed that the kinetics of damage sensitivity induced by organic hydroperoxides compared to hydrogen peroxide differs over time. Peroxide produced an initial burst of bacterial killing that then remained at similar levels after longer incubation times, whereas organic hydroperoxides continued to induce increased damage sensitivity over longer time periods (Supplementary Fig. 8), probably reflecting their stability in cultures.

Prim-PolC and LigC1 were previously reported to contribute to DNA repair during stationary phase in *M. smegmatis*[12, 14]. Loss of Prim-Pol (Δ*prim-polC*) or ligase (Δ*ligC1, C2*) components of the LigC complex caused a similar level of impaired survival when treated with either of these oxidising agents (Fig. 6b, Supplementary Fig. 8). As expected, PolA deficient strains were the most sensitive to these genotoxins. Cumene hydroperoxide is among the best known inducers of PolA expression in mycobacteria, suggesting that it is one of the key enzymes necessary for the repair of DNA lesions caused by organic hydroperoxides. Unexpectedly, LigD-deficient strains were even more sensitive to these oxidising agents and simultaneous loss of LigD and Prim-PolC (Δ*ligD, prim-polC*) had an additive impact on cell survival (Fig. 6b, Supplementary Fig. 8), indicating that both pathways are important for repairing oxidative lesions in stationary phase.

## Discussion

Prior to this study, it was proposed that the LigC complex functions as an alternative "back-up" pathway[8, 14] required for repairing DSBs in stationary phase, following the loss of the Ligase D complex[5, 29]. However, this hypothesis has not been experimentally proven and therefore the cellular function(s) of this pathway remained unclear. A major issue hampering the assignment of a biological role for the LigC complex, distinct from LigD, has been the almost indistinguishable biochemical activities shared between these closely related repair complexes. These include an affinity towards 5′-P moieties on their respective DNA substrates and a preference to incorporate short patches of ribonucleotides to fill in gapped intermediates, followed by requisite resealing by a specific RNA/DNA ligase activity[24, 25]. These similarities also extend to the structures of Prim-PolC and D, which are highly superposable and share similar prominent surface loops (Loop 1 and 2) and a 5′ phosphate binding pocket that together dictate the preferential substrate binding specificity of Prim-PolD for DNA overhangs and break-annealed gaps[23–26].

Despite these similarities, this study reveals that Prim-PolC possesses a number of distinguishing biochemical and structural features that enable us to define its preferred DNA substrates. Prim-PolC preferentially binds to and fills-in short DNA gaps. Unlike Prim-PolD, it shows weak preference for overhangs and is unable to mediate break synapsis and synthesis. It also contains a

conserved C-terminal structural element (Loop 3), which is absent from Prim-PolD orthologues. This loop is positioned between loops 1 and 2, where it forms a continuous surface that is probably involved in directing the template strand towards the active site to optimally position, with Loops 1 and 2, the 3′ hydroxyl moiety of the primer strand into the active site in preparation for extension. The absence of Loop 3 in Prim-PolD orthologues, leaves a space between the remaining loops (1 and 2), which allows these NHEJ polymerases to promote break annealing, MMEJ and processing of subsequently annealed DSBs[23–26]. Validation of this proposed role for Loop 3 awaits the elucidation of additional structures of Prim-PolC bound to DNA gap intermediates.

In addition to these molecular analyses, our in vivo studies establish that the LigC complex is a central nexus for a distinctive excision repair pathway required to remove and replace damaged or modified bases in mycobacterial genomes during stationary phase (Fig. 6c). BER reconstitution studies in vitro demonstrated that LigC-dependent excision repair complexes can repair oxidatively damaged bases and probably other lesions, e.g., methylated/nitrosative damaged bases and SSBs, that are all processed via an abasic site intermediate. A key feature of this repair pathway is the preferential incorporation of ribonucleotides, consistent with their abundance in stationary phase. We observed the most efficient repair in vitro when endonuclease IV was employed as the gap processing enzyme, dephosphorylating the 3′-phosphate end of the lesion resulting from β,γ-elimination of the abasic site, catalysed by FPG. EndoIV lacks a strong 3′-5′ exonuclease activity, present in alternative end-processing enzymes (ExoIII and XthA), and therefore this appears to limit the gap resection to an optimal size for LigC-mediated repair. Consistent with this, it was previously reported that EndoIV, rather than ExoIII, is responsible for vast majority of BER gaps processing in mycobacteria, in contrast to eubacteria where ExoIII plays a more dominant role[30].

In mycobacteria, the expression of the LigC complex peaks during early stationary phase. This coincides with a significant reduction in intercellular pools of dNTPs in the absence of replication[31, 32], probably explaining the necessity to incorporate ribonucleotides during DNA repair processes in stationary phase. Another notable feature of the LigC pathway is the specific incorporation of short ribonucleotide patches that are likely resistant to RNase H1 activities[33–35]. Apart from Prim-PolC/D, DinB2 also incorporates ribonucleotides into DNA during stationary phase repair processes in mycobacteria[36]. However, this Y-family polymerase has a tendency to incorporate longer RNA patches and the resulting DNA:RNA hybrids would be sensitive to RNase H1 digestion. In gap-filling assays, Prim-PolC efficiently filled in short gaps (1–3 nt) but failed to process longer gaps (5 nt). Notably, stretches of >4 ribonucleotides incorporated into dsDNA are known to exhibit high instability due to their sensitivity to RNase H1 digestion in vivo[33–35]. It is therefore likely that short RNA patches incorporated by Prim-Pols during BER and NHEJ repair processing are later recognised and replaced by PolA, rather than RNase H, which undertakes a similar role during RNA primer removal, once intracellular pools of dNTPs are restored[37, 38].

Prim-Pols have relatively low fidelity during synthesis and therefore another reason to favour the incorporation of ribonucleotides during repair is to "mark" these regions of repair synthesis for subsequent replacement in a post-repair manner by more accurate polymerases[5, 10], reminiscent of RNA primer removal during replication. PolA removes RNA primers and is also known to participate in several DNA repair pathways in bacteria[37, 39–41]. PolA-driven DNA synthesis occurs slowly but accurately and it has proofreading and bidirectional nuclease

activities beneficial for correcting errors introduced by Prim-Pols[39, 42]. Notably, in this study, we also detected a weak association between PolA and the LigC complex, suggesting that this replicase may remove and replace RNA inserted by this pathway, and likely others e.g. NHEJ, with more accurately synthesised DNA.

Both Δprim-polC and ΔligC1,C2 strains were similarly sensitive to oxidising agents, consistent with their proposed joint roles in a distinctive excision repair pathway in mycobacteria. Surprisingly, a LigD-deficient strain was even more sensitive to these agents, although this could potentially be attributed to a deficit in DSB repair. However, this sensitivity was evident even with low levels of peroxide (~5 mM), reported to produce predominantly SSBs and <0.1% DSBs[43], thus uncovering an unexpected and additional role for LigD in excision repair. Consistent with this proposal, co-deletion of LigD and Prim-PolC had an additive impact on cell survival in the presence of these genotoxins, also indicating that they are not epistatic in their function. Consistent with the mutant phenotypes, both LigD and Prim-PolC co-purified on a 5′-phosphorylated single nucleotide gap DNA substrate in pull-down assays. This substrate mimics a naturally occurring BER DNA intermediate that would arise from the enzymatic processing of an oxidative lesion by FPG and EndoIV. Together, these findings establish that both LigC and LigD-dependent complexes contribute significantly to the repair of lesions produced by oxidative damage during stationary phase in mycobacteria. Notably, Ory et al.[44] reported recently that Bacillus subtilis LigD possesses dRP-ase activity in vitro supporting this proposed role in excision repair. Further studies are required to define the exact nature of the oxidative lesions that these Prim-Pol pathways process in vivo and uncover the overlap and "crosstalk" that occurs between these, and other, stationary phase DNA repair pathways in mycobacteria.

## Methods

**Bacterial strains and growth conditions**. Laboratory stock of M. smegmatis mc[2] 155 and its derivatives were cultured in 7H9 liquid broth or 7H10 solid medium (Beckton Dickinson) with ADC supplement (Albumin-Dextrose-Catalase) and hygromycin B (50 μg ml[−1]) or kanamycin (30 μg ml[−1]), if antibiotic selection was required. Escherichia coli DH5α strain (Invitrogen) was used for cloning purposes and Origami 2(DE3)pLysS (Novagen) for protein purification, respectively. E. coli strains were cultured in standard Luria broth medium or terrific broth (Formedium), where large amounts of protein production was required with addition of kanamycin (50 μg ml[−1]) for selection.

**Protein complex purification and mass spectrometry analysis**. Genes encoding Prim-PolC, LigC1 and LigC2 were PCR amplified and sub-cloned into pKW08-gfp vector using a sequence and ligation independent protocol to produce C-terminal eGFP fusion proteins used as baits for protein purification[15]. Purification and data analysis was performed essentially as described previously[15]. Briefly, mycobacterial cells were collected by centrifugation (15 min, 4800×g, 4 °C) and resuspended in a cold sonication buffer containing 50 mM Tris (pH 8.0), 100 mM NaCl, 1 mM dithiotreitol, 2 mM phenylmethylsulfonyl fluoride, 25 U ml[−1] benzonase, 0.5% Triton X-100 and protease inhibitor cocktail. The cells were lysed by sonication and lysates were pre-cleared by centrifugation (20 min, 4800×g, 4 °C). Lysates were then mixed with 50 μls of the anti-GFP Sepharose beads (Chromotek), incubated for 2 h at 4 °C to capture protein complexes, washed two times with 10 ml of wash buffer (10 mM Tris (pH 8.0), 150 mM NaCl and 0.1% Triton X-100), followed by two washes with TEV buffer (10 mM Tris (pH 8.0), 150 mM NaCl, 0.5 mM EDTA and 1 mM DTT) and digested with TEV protease (Thermo Fisher Scientific) to elute protein complexes.

For DNA-protein pull-downs, each DNA substrate at a final concentration of 50 nM was annealed to 1 mg of the poly-dT magnetic particles (NEB) in buffer containing 20 mM Tris-HCl, pH 8.0, 10 mM MgCl₂, 0.1 M KCl, for 3 h at room temperature and washed twice with the wash buffer (50 mM HEPES pH 7.5, 150 mM NaCl, 0.5% Triton X, 1 mM EDTA). Whole-cell lysates were prepared as described above, however, in 50 mM HEPES pH 7.5 as buffering agent and without the addition of nucleases. The pre-cleared lysates were filtered through 0.22 μm filters and incubated with the DNA coated beads for 3 h at 4 °C. For the last 10 min 2 μls of RNaseA (10 mg ml[−1]) were added to the 1 ml reaction. Beads were collected on the magnetic stand and washed five times with the wash buffer. DNA bound

protein complexes were then eluted by DNaseI digestion (NEB) for 1 h at 37 °C in dedicated buffer.

The resultant protein mixture was precipitated with pyrogallol red–molybdate (PRM; 0.05 mM pyrogallol red, 0.16 mM sodium molybdate, 1 mM sodium oxalate, 50 mM succinic acid, pH 2.5), collected by centrifugation and subjected to a standard trypsin digestion. The resulting peptide mixtures were applied to a nano-HPLC RP-18 column (Waters) using an acetonitryle gradient in the presence of trifluoroacetic acid. The column was directly linked to the ion source and the Orbitrap (Thermo Scientific) was operated in a data-dependent mode.

The MaxQuant (v1.3.0.5) computational proteomics platform was used to process raw MS files. Integrated Andromeda search engine and appropriate *M. smegmatis* or *M. bovis BCG* protein databases were used to search against the fragmentation spectra. Carbamidomethylation of cysteines, N-terminal acetylation and oxidations were set as possible peptide modifications. One per cent false discovery rate was applied to all protein and peptide identifications. Human derived contaminants and random protein identifications were excluded from the results files. Proteins considered as valid identifications were identified by at least two peptides.

**Purification of multiple DNA repair proteins**. Genes encoding Prim-PolC, PolD2, LigC1, LigC2, MPG, FPG, NTH, EndoIV, ExoIII, XthA, XseVIIB and PolA were PCR amplified using primers flanked with restriction digestion sites required for in frame cloning into pET28 vector (Supplementary Table 1). All the proteins were designed to contain a N-terminal His-tag, except for FPG, which was rendered inactive unless tagged at its C terminus. Proteins were purified according to routine laboratory procedures using ÄKTA purifier and compatible columns purchased from GE healthcare. Briefly, pre-cleared cell lysates obtained after overexpression of recombinant proteins in *E. coli* Origami B pLysS strain were loaded onto Ni Sepharose (Qiagen) column in Tris buffers (50 mM Tris pH 8.0), washed extensively and eluted in a gradient of imidazole. Proteins were next loaded onto an ion exchange column (SP or Q (GE Healthcare), depending on the iso-electric point (pI)) eluted in a gradient of NaCl and further purified on preparative gel filtration columns (S200 or S75 (GE Healthcare), depending on the molecular mass of purified protein). Quality of proteins after each and every purification step was checked using sodium dodecyl sulfate (SDS) polyacrylamide gel electrophoresis (PAGE).

**EMSA**. Assays were carried out essentially as described previously[10, 25]. Briefly, EMSAs were employed to determine optimal substrates for Prim-PolC activity and were set up in 50 mM Tris-HCl (pH 7.5), 0.1 mg ml⁻¹ of BSA (NEB Biolabs), 1 mM DTT, 5% glycerol, with indicated 30 nM 6-carboxyfluorescein (6-FAM) labelled DNA substrates and different concentrations of Prim-PolC in a 15 µl reaction volume. Incubation mixtures were kept on ice for 30 min and were resolved by native gel electrophoresis on a 5% polyacrylamide gel (80:1 (w/w) acrylamide/bisacrylamide).

**Polymerisation and DNA repair assays**. DNA extension reaction mixture contained 50 mM Tris-HCl (pH 7.5), 5 mM MgCl₂, 100 µM MnCl₂, 30 nM 6-FAM labelled DNA substrate, 250 µM NTPs and indicated DNA repair proteins, in a total volume of 20 µl. Reactions were further supplemented with 1 mM ATP, 1 mM DTT and 0.1 mg ml⁻¹ bovine serum albumin (BSA) for DNA repair reconstitution reactions. After a set incubation time at 37 °C, reactions were terminated by adding stop buffer solution (95% (v/v) formamide, 0.09% (w/v) bromophenol blue, 20 mM EDTA and 600 nM unlabelled oligonucleotide, identical to labelled strand). Resulting DNA extension or repair products were resolved for 2 h at constant wattage of 20 W, on TBE-buffered 15% polyacrylamide gels containing 8 M urea. Detection of fluorescently labelled oligonucleotide products was carried out using Fujifilm FLA-5100 fluorescent image scanner and EMSA data were quantified using Quantity One (Bio-Rad).

**Antibodies and western blotting**. Antibodies were generated against *M. smegmatis* proteins (Ku, LigC1 or Prim-PolC) using a 28 day rabbit immunisation program (Eurogentec), using ~1 mg full-length recombinant protein (two applications), with serum collected at different time points. Recombinant proteins (5 µg per lane) were resolved by 10% SDS-PAGE and proteins transfer onto polyvinylidene fluoride (PVDF) membrane, which was subsequently blocked with Tris-buffered saline and Tween 20 (TBST) and 5% (w/v) non-fat dried milk. Membrane bands containing the recombinant protein were excised, washed twice in TBST and incubated overnight at 4 °C with 500 µl of serum. Serum was next removed and membranes washed (x4) with 500 µl of TBST. Purified immunoglobulins bound to the membranes were eluted with 150 µl of glycine buffer (500 mM NaCl, 50 mM glycine, 0.5% Tween 20, 0.1% BSA, pH 2–3) and the blot vortex pulsed before adding 25 µl of 1 M Tris pH 8.8.

For western blotting analysis, the primary antibodies (anti- Ku, Prim-PolC and LigC1) were used at a dilution of 1/1000 and the secondary antibody (HRP-conjugated Anti-Rabbit IgG; Abcam ab6721) at a dilution of 1:5000 dilution. Uncropped versions of all western blots and gels can be found in Supplementary Fig. 9.

**Far-western analysis**. This method was used to test for protein–protein interactions in vitro. Recombinant proteins were first slot-blotted using a slot blot manifold (GE Healthcare) onto a methanol-wetted PVDF membrane according to the manufacturer's instructions. Typically, a fixed concentration of recombinant protein (50 ng for the probed protein and 3 µg for the possible interactors) was blotted onto the membrane. The membrane was first blocked with Tris-buffered saline (TBS: 280 mM NaCl, 20 mM Tris) containing 0.05% (v/v) Tween 20 and 5% (w/v) non-fat dried milk. For far-western analysis, the membrane was then incubated with blocking buffer containing 5 µg ml⁻¹ of the candidate interacting recombinant protein. The membrane was then washed, probed with primary (1/1000 dilution) and secondary (1/5000 dilution) antibodies, and subjected to chemiluminescent detection. The primary antibody used was specific for the candidate interacting recombinant protein, and chemiluminescence detected if interaction occurred between the two proteins.

**Pull-down assays**. Pull-down assays were used to confirm chosen protein–protein interactions. Equimolar amounts of all proteins were used at 5 nM per pull-down. Recombinant FPG-6xHis or 6xHis-MPG bait proteins were first bound to nickel-nitrilotriacetic acid (NTA) coupled resin in a Tris-based buffer (50 mM, pH 8.0) containing 150 mM NaCl, for 1 h at 4 °C. The protein-coated beads were washed and interacted with tag-free LigC (6xHis-LigC subjected to a standard thrombin digestion) or BSA, for further 2 h. The beads were next washed extensively and protein complexes were eluted with buffer containing 300 nM imidazole. The protein complexes were visualised by SDS-PAGE followed by Coomassie staining.

**Crystallisation and X-ray structure determination**. Crystals of the apo Prim-PolC were grown at 285 K by vapour diffusion as sitting drops. The protein was screened at a concentration of 8 mg ml⁻¹ and 0.5 µL of protein solution was mixed with 0.5 µL of crystallisation buffer (0.05 M Tris-HCl (pH 8.0), 0.2 M ammonium chloride, 0.01 M calcium chloride, 30% (w/v) PEG 4000). Prior to data collection, crystals were soaked in mother liquor containing 20% ethylene glycol. 0.9795 Å X-ray diffraction data were collected at 100 K using a synchrotron source at station I02 Diamond Light Source, Didcot, UK. The diffraction data were processed with xia2[45] with additional processing by programs from the CCP4 suite (Collaborative Computational Project, Number 4, 1994). The statistics for data processing are summarised in Table 1. Initial phases were obtained by molecular replacement with PHASER[46] using Prim-PolD (2IRY) as a search model[23]. Iterative cycles of model building and refinement were performed using Coot[47] and Phenix[48]. A final refined model at 1.84 Å resolution, with an $R_{factor}$ of 18.58% and $R_{free}$ of 20.92% was obtained. 99.1% of residues are in preferred regions with the remainder (0.9%) in allowed regions according to Ramachandran statistics. Structural images were prepared with CCP 4 mg[49]. The structure of *apo* Prim-PolC is deposited in the Protein Data Bank under accession code 5OP0.

**Construction of mutant strains lacking DNA repair genes**. Targeted gene replacement strategy was employed to generate unmarked ∆prim-polC, ∆ligC1, ∆ligC2, ∆ligD and ∆polA:: PolA1-755aa mutants and strains with simultaneous removal of multiple of these genes as previously published[50, 51]. Briefly, short fragments corresponding to 3′ and 5′ end of each genes, flanked with surrounding genomic region of about 1 kb at each side (primers listed in Supplementary Table 1), were ligated together out of frame and sub-cloned into a suicidal p2NIL delivery vector, merged with crossing over event selection cassette from pGOAL17. Multistep screening led to double crossing over strains selection and resulted in gene inactivation, which was confirmed by Southern blotting and PCR, for each recombination event.

**Phenotypic analysis of *M. smegmatis* strains**. *M. smegmatis* MC2 155 wild-type (ATCC 700084) and mutant strains were grown to late stationary phase in 7H9 liquid media, with supplements, typically 4 days after reaching optical density (OD 600 nm) of 3.0. The cells were collected and suspended in PBS containing Tween 80 (0.05% (v/v)), to prevent clumping, at OD 600 of 0.1. They were next treated with 10 mM cumene hydroperoxide (from 1 M stock in 50% ethanol), 50 mM tert-butyl hydroperoxide (from 5 M stock in water), or 5 mM hydrogen peroxide (from 0.5 M stock in water), respectively, for various amount of time with incubation at 37 °C. Untreated cells suspension was used to control overall viability of each strain. At indicated time points 100 µls of each cell suspension (diluted ten fold) was transferred onto 7H10 solid agar and plates were incubated for 72 h before read out of cell survival was recorded.

**Data availability**. All data are provided in full in the results section and the Supplementary Information accompanying this paper, or from the authors upon request. The crystal structure has been deposited in the Protein Data Bank and can be accessed using accession code 5OP0.

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

## Acknowledgements

We thank Dr M. Roe for assistance with X-ray data collection. A.J.D.'s laboratory was supported by grants from Biotechnology and Biological Sciences Research Council (BB/F013795/1, BB/J018643/1 and BB/M004236/1). P.P. was supported by a mobility grant from the Polish Ministry of Science and Higher Education (1073/MOB/2013/0: Mobilnosc Plus). Funding for open access charge: Research Councils UK (RCUK).

## Author contributions

A.J.D. designed the project, directed the experimental work, performed database analysis and wrote the manuscript. P.P. and N.C.B. contributed to the project design, performed and designed experiments, performed computational analyses and co-wrote the manuscript. P.P. performed mass spectrometry analysis. N.C.B. performed X-ray data processing, model building and analysis. J.B. generated LigC and Prim-PolC expression plasmids, purified proteins, measured protein expression levels in cells and performed the initial biochemical and phenotype studies. A.B. and M.K.-M. constructed mutant strains in *M. smegmatis* and assisted P.P. with CFU experiments. A.D. and J.D. helped with study design and directed parts of the experimental work (mass spectrometry analysis of protein complexes and generation of mutant strains and phenotypic analysis, respectively).

## Additional information

**Competing interests:** The authors declare no competing financial interests.

