## [Peer Review File · Nature Communications]

Reviewers' Comments:

Reviewer #1 (Remarks to the Author)

In this manuscript, the authors characterize the functions of the LigC1/PrimPolC complex in *Mycobacterium* spp., showing that it physically and functionally interacts with the BER machinery, facilitating repair of DNA base lesions. Whether this pathway is also operating in organisms other than *Mycobacterium* spp., which show orthologs of LigC has not been addressed. The manuscript shows some weaknesses, as listed below.

MAJOR POINTS

- 1) Figure 2. The results of the far-western analysis should be confirmed by co-IPs or pulldowns for at least MPG and FPG, also to directly prove that LigC is required to have the ternary complex with PrimPolC.
- 2) Figure 3a. The expression "dramatically stimulated" is quite general. Quantification of the EMSAs is required. Explain the arrows in Fig.3a. The legend of Figure 3a states that fixed amounts of proteins were used, while the figure suggests that a titration of PrimPolC was performed. Please state the concentrations. Why two molecular species are detected in the right panel? If 300nM was the highest concentration, why the shifted signal in Fig.3a is different from the one in FigS2a obtained with a fixed concentration of 300nM?
- 3) Figure 3b and 3d. When stated that it was "observed a preference of CTP insertion", Fig. 3 b is indicated in the text. However, Fig. 3b does not address the fidelity (all 4 rNTPs are present), thus here it should be cited Fig. 3d instead. Fig.3b shows that rNTPs are better incorporated when the 5'-end is not phosphorylated, while the difference with phosphorylated template is much less pronounced. There is more unextended product with dNTPs, but also the extended products look more, as if the total amount of template was not equal between rNTPs and dNTPs experiments. If the authors want to make this as a strong point, a nucleotide titration is required in order to quantitatively assess the difference. Was the synthesis due to CMP incorporation only or was it dependent on the template sequence? The experiment shown in Fig.3d clearly indicated that with 5'phospho ends all four rNTPs and dNTPs can be incorporated opposite a template T, and it is hard to say that CMP is preferred without proper quantification. In general, one of the major authors' conclusion is that the LigC complex preferentially carries BER using rNTPs (cited also in the discussion, p.15). However, such clear preference was not demonstrated. The reactions were not carried out with physiological levels of either rNTPs or dNTPs (which are not equimolar mixtures). In addition, no selectivity values are given, making it impossible to state what the real preference is and if under physiological conditions rNTPs may really compete with dNTPs for incorporation. A competition assay including rNTPs and dNTPs at physiologically relevant ratios should be carried out, at least in an experimental setup similar to Fig. 3d, to show the relative incorporation (the products can be distinguished by their different electrophoretic mobility). In addition, selectivity indexes should be derived for PrimPolC rNMPs vs. dNMPs incorporation.
- 4) Figure 3c. The ligated products appear heterogeneous, with at least two or three molecular species migrating with decreasing mobility. If the band representing the expected product is the major one, were the other species due to extra added nucleotides? If so, what is the explanation? Does it mean that the repair with rNTPs is error-prone (low fidelity of insertion plus extra nucleotide addition?)
- 5) Figure 6. Why deletion of ligC1 and C2 gives better tolerance to oxidative stress than deletion of ligC2? Why ligC1 (which was the main protein studied here) was not deleted alone? If ligC1 works together with PrimPolC, one should expect to observe the same increased sensitivity as the PrimPolC deleted mutant for a ligC1 deletion. The data, however, seem to indicate that PrimPolC is important for oxidative stress tolerance irrespectively from ligC1/C2 (compare the effects of ligC1/C2 deletion with the PrimPolC alone or the triple deletion).
- 6) In the text it is not clarified whether the authors also carried out mock-purification controls for their eGFP fusion tags, in order to exclude unspecific protein binding. This should be specified.

MINOR POINTS

- 1) Introduction, p.2 line 9 from top: "is...forms" should be "is...that forms"; p.3, line 5 from bottom: RNA is not incorporated, rNTPs (or even better rNMPs) are. Replace RNA with rNTPs.
- 2) Results, p.5 line 4 from top: some detail about the nature of the C-ter extensions (or some relevant reference) may be of interest. p.5, line 7 from bottom: please define DSSP
- 3) Figure 2a: in the Table there is a problem with the character encoding in the PDF in the row of uvrB
- 4) p.6: EndoIV results may be shown as supplementary data
- 5) p.7, after "genetic interactions" Figure 2c should be cited, since it is not cited anywhere else.
- 6) Fig. 3b. The size markers on the right side of panel b are shifted up

Reviewer #2 (Remarks to the Author)

This work is focused on the DNA repair apparatus called Ligase C complex. The authors showed that the LigC complex is involved in base-excision repair (BEC) in mycobacteria. Moreover, their studies indicate that LigC complex interacts with BER proteins *in vivo* and together form an apparatus with a distinctive function in DNA repair (i.e. removing and mending damaged bases and abasic sites). Moreover, they show that cells lacking the LigC complex are more sensitive to oxidative DNA damage.

In the current state, this work does not seem suitable for this journal. Perhaps some major revisions may help.

1. The writing style is rather difficult to follow, probably addressed to specialists rather than to a broad audience. The names of the many proteins need to be introduced in a more structured and uniform manner. For instance, they refer to "Ligase C", sometimes "LigC", sometimes to "LigC complex". It is not easy to figure out if these are alternative names for the same entity, or whether they refer to a multiprotein complex assembled on a scaffold-like subunit enzyme called LigC. The difference should be clarified right from the beginning, since this refers to key-terms of the manuscript.

Another example of ambiguity refers to the first paragraph of the "Results" section ("Actinobacteria possess multiple.."). In the way how the text is described, it is not clear if this represents introductory background information, or the result of their own bioinformatics analyses.

In connection to this chapter, it is not clear the relationship between the proteins depicted as arrows of different colors in Fig. 1. Do they act in the same pathway? Are they also part of the same complex/machinery? Probably the figure is supposed to be self-explanatory.

2. The proteins Prim-PolC, LigC1 and LigC2 were co-purified as baits with multiple proteins involved in the BER pathway. The complexes were cross-linked with DSSP and subjected to mass-spectrometric analysis for the identification of the captured proteins.

However, some important questions are not addressed. For instance, are all these listed proteins forming a single complex? Or they reflect various smaller subcomplexes that bind the same bait? To verify this aspect, one would need to check the migration of the affinity purified complexes, prior to cross-linking and mass-spec (for instance by native PAGE or gradient ultracentrifugation). Perhaps an SDS-PAGE of the affinity-purified complex should be also shown.

3. Next, the authors use recombinant proteins to validate the interactions detected between bait proteins (LigC1 and Prim-PolC) and BER proteins. The interactions were evidenced by western-blotting with specific antibodies (Fig. 2b). They conclude that LigC1 acts as a key scaffolding protein required for sequential recruitment of BER proteins.

Giving the availability of the recombinant proteins, some additional, perhaps very informative

experiments can be performed. For instance, they can evidence/describe the interactions by other methods, as size-exclusion chromatography), which additionally provides information about the stability of the complexes. This may also answer to questions like: (i) do all these proteins form a single machinery, that behaves as a single particle? (ii) alternatively, do these proteins associate in several distinct complexes? If yes, which proteins are associated physically? (iii) Do the proteins act transiently, one after another in a step-wise manner? If so, perhaps some competition experiments can be performed to support the description of the pathway where they act. Perhaps the authors regard some of these experiments unnecessary, based on background information from the DNA repair field, and/or parallels with the LigD action. However, if this is the case, the authors need to discuss this convincingly.

4. The authors solved the crystal structure of Prim-PolC and compared it to the previously reported Prim-PolD. The two structures are very similar, except for an additional C-terminal stretch that is present only in Prim-PolC, and highly conserved among its orthologues. This element (called Loop3) forms a layer with the canonical loops 1 and 2, on one side of the active site of the enzyme.

From a technical standpoint, the structure is absolutely fine. However, the importance of loop3's location is over interpreted and unconvincing in the absence of supporting experiments. Moreover, it is not conceptually well-linked to the flow of their story. Probably for this reason the authors do not mention the crystal structure in the abstract of the manuscript.

One important question is whether loop3 is folded in the way it is due to the crystal contacts. Even assuming that it is not involved in contacts, one should assay the activity of the enzyme in the absence of this loop. Moreover, point-directed mutagenesis will be required to support the rather detailed interpretation of the structure.

Reviewer #3 (Remarks to the Author)

Plocinski et al. present a study of prokaryotic/mycobacterial LigC, an ATP-dependent DNA repair ligase. The Doherty group and others have previously done substantial work on a related protein, LigD, and shown it to comprise a multifunctional nonhomologous end joining (NHEJ) double-strand break (DSB) repair complex active in stationary phase. In contrast to prior descriptions that LigC might function in a redundant NHEJ pathway, the current study provides evidence from a number of different experimental approaches that LigC acts as part of a base excision repair (BER) single-strand break (SSB) pathway, again most active in stationary phase. Similar to LigD, LigC incorporates rNTPs, consistent with function in a non-replicative phase when dNTPs are limiting.

The LigC, PrimPolC and related proteins are of considerable interest in important microbiology and as correlates to other species, as well as revealing of general principles in DNA repair. As such, a definitive description of a dedicated role of LigC in a unique BER pathway would be highly significant as well as novel. A strength of this study is the number of different approaches used, even including crystallography to describe a novel protein loop. The result is a collection of supportive findings that create a plausible and compelling model of LigC function. Unfortunately, in being so diverse, many specific aspects of the study are not performed at enough depth to make them as compelling or informative as needed, so that in the end the study falls short of its claim of definitively "[establishing] a distinct DNA repair pathway required to excise and replace ... bases ... during stationary phase".

Specific points:

1) The 2014 J. Bacteriology paper from Glickman is a clearly a central reference point for this study, as it claimed a role for LigC in NHEJ. First, it would seem relevant to reference this paper at the appropriate place in the 2nd Introduction paragraph. The Glickman study also measured LigC expression as a function of culture growth stage – how is the current study additionally informative? Most importantly, the Discussion fails to ever consider the current results in the context of the prior study, simply saying that the NHEJ connection was "not proven", which is not

helpful to readers.

2) At many points, the text is very rough and in need of grammatical correction. There are too many instances to list, but a first instance can be found in the first Introduction paragraph "The best characterized AEP is PolDom forms part of a ..."

3) Figure 1 is of dubious value to the paper, and problematic in its presentation. First, is it a novel description (it seems not)? If essential to rapid understanding of the current study, it is failing that purpose as the labeling is inadequate – why wouldn't the genes be labeled with their identities as text? Are all species essential to display? Is the disposition within the genome critical for understanding this paper? Finally, the '&' symbol is meaningless without consulting the legend, which shouldn't be necessary for such a simple figure.

4) Extending the previous point, the first Results paragraph has a lot of descriptive detail that does not seem novel or essential. The point is unclear and being lost. I think it could be substantially shortened and moved to Introduction, as the purpose seems to be helping the reader understand that LigC is a structurally definable and broadly conserved protein, previously established information.

5) Can Fig. 2a include gene names for all entries? It is unnecessarily difficult to identify all proteins (e.g. "Deoxyribonuclease" is uninformative).

6) The slot blotting method in Figure 2b is useful as a rapid way of probing many potential interactions, but is not quantitative and subject to a number of assumptions. Findings would be more compelling if confirmed individually by pull-down types of experiments. Also, while the Ku control is important, it might be useful to include LigD, given previous finding that LigC can cooperate during NHEJ.

7) In Figure 3a, the arrows and bands are never defined. I gather the presumption is that the black arrow denotes a specific complex at the nick/gap, and the grey arrow denotes a non-specific complex; has this been validated?

8) Figure 3b is difficult to understand. First, it is lacking a critical no-enzyme control. Second, the 0/+1/etc. labels do not appear to line up with the correct bands, and are very distant from the leftmost lanes in any case. I was ultimately able to convince myself that the gel is understandable if the enzyme adds 1 nt to a nick, but it took me too long to get there. Finally, the depiction of the overhang/ov substrate is not intuitive; I'm still not certain what it is.

9) Figure S2b lacks any lanes to establish the position of the expected product from the DSB ligation substrates. Among other things, this could include incubation with LigD. Also, these lanes never seem to be mentioned anywhere – while they aren't definitive proof of a lack of DSB ligation ability of LigC, or of a preference for nicks (given that any ligase can ligate a nick), it seems relevant to mention.

10) Figure 3d is never mentioned and without discussion is difficult to understand. I think perhaps it was intended to be omitted and is related to a comment at an earlier point in the text? I agree with omitting it, it doesn't strongly support the most important points of the paper.

11) In Figure 4b the authors have tried to be too clever in their labeling and made it harder, not easier, to understand what ultimately is a conceptually simple experiment, that ligated products can be seen when all required lesion processing activities are added. To the extent that I believe I ultimately understand the experiment, it cannot be considered revealing beyond what is known about their individual enzymatic activities. In the absence of more careful studies of combinations, order of addition, amounts, kinetics etc. we learn little additional about the enzymes, and of the functional interactive complex which is being claimed. Even the simplest controls like lanes

omitting LigC are absent. And, what if LigD were added instead of LigC, i.e. how specific is this to LigC?

12) The structural studies of LigC are the most compelling part of the manuscript and very interesting. I think the very long paragraph describing the loops should be broken up to make the different important insights more digestible. Figure 5a seems to have lost a "Loop 3" label that would be helpful to add back. Also, the authors never actually define Loop 3 in the text – they just start using it at one point. Labeling of the 5' phosphate and binding pocket will also help orient readers, as would labeling the inferred DNA:loop 3 steric conflict. Ultimately, though, while the interesting structure of LigC loop 3 almost certainly does alter its catalytic behavior, insufficient data are provided to develop a binding model for a nick/gap substrate – things are currently largely conjectural.

13) If Figure 6b, it is never clearly stated why TBH and CHP were used, as opposed to other agents (H₂O₂?) -is there something important about organic oxidizing agents? Also, I find the text description of the results very confusing and difficult to match up to the figure. To me, the text implies construction of strains that are not shown (e.g. a combined PolD2 Prim-PolC mutant). It is also unclear and not addressed why ligC2 alone seems to have a larger effect than ligC1,ligC2 combined. Also, why is PrimPolC more sensitive than ligC1,ligC2? The rest of the paper might give the impression that LigC is the defining protein of the pathway. Finally, to support the claims of the paper it would be important for comparisons of sensitivity to be made at different growth stages.

14) Also regarding the sensitivity to oxidative damaging agents, while there are certainly single-strand lesions there may also be double strand lesions, so it can be difficult to be certain which lesions are responsible for LigC mutant sensitivity. Further study of genetic interactions would help interpretation, including with LigD and well as with other SSB repair components, where different outcomes of synergy and epistasis might be expected. In other words, can it be bolstered that the in vivo phenotypes are truly due to SSB, not DSB repair?

15) The discussion of why it makes biological sense to have an RNA-incorporating BER pathway in stationary phase is clearly relevant and important to the Discussion but quite long; I think this could be tightened up considerably. It seems to have been engaged at the exclusion of consideration of any of the structural biology, or of a clear comparison of current to previous studies.

Reply to reviewers' comments:

We would like to thank all the reviewers for their most insightful ,helpful and positive comments on our manuscript, which we have taken full advantage of during the revision process. We have worked hard in the intervening time to address your major concerns, including many significant experimental additions that are described below in our point by point response. We believe that these revisions and additional data significantly enhance the interpretation, focus and impact of our paper.

Reviewer #1 (Remarks to the Author):

In this manuscript, the authors characterize the functions of the LigC1/PrimPolC complex in *Mycobacterium* spp., showing that it physically and functionally interacts with the BER machinery, facilitating repair of DNA base lesions. Whether this pathway is also operating in organisms other than *Mycobacterium* spp., which show orthologs of LigC has not been addressed. The manuscript shows some weaknesses, as listed below.

MAJOR POINTS

1) Figure 2. The results of the far-western analysis should be confirmed by co-IPs or pulldowns for at least MPG and FPG, also to directly prove that LigC is required to have the ternary complex with PrimPolC.

Co-IPs have now been performed with His tagged MPG and FPG and tag-free LigC1, showing a strong interaction between LigC and FPG (Supplementary Figure 3b). We therefore have two independent methods showing an interaction between FPG and LigC, which together with the genetic / operonic association of these proteins, makes the connection between the two enzymes highly significant.

2) Figure 3a. The expression "dramatically stimulated" is quite general. Quantification of the EMSAs is required. Explain the arrows in Fig.3a. The legend of Figure 3a states that fixed amounts of proteins were used, while the figure suggests that a titration of PrimPolC was performed. Please state the concentrations. Why two molecular species are detected in the right panel? If 300nM was the highest concentration, why the shifted signal in Fig.3a is different from the one in FigS2a obtained with a fixed concentration of 300nM?

The titration of Prim-PolC binding to gapped substrate is represented by the EMSA in Figure 2a (formerly Figure 3a). The concentrations of Prim-PolC used were 0, 12.5nM, 25nM, 100nM, 250nM, 500nM, 1 μ M, 5 μ M 10 μ M. In our lab, we only use EMSA's for the qualitative assessment of binding due to the inherent variation in this method. As the gels in Figure 2a were run at the same time with the same protein sample, a direct qualitative assessment can be made in this case. We consider that "dramatically stimulated" is appropriate in this case, as virtually all the phosphorylated probe is shifted by Prim-PolC concentrations above 100nM. We believe that quantification is not required as this is only a qualitative comparison, no more, and there is an obvious large difference of shifted probe at the concentration range shown.

The legend has now been amended to clear up the confusion and an explanation of the arrows is included, "A filled triangle indicates unshifted probe, whereas the black arrow

probably indicates a binary complex of PrimPoIC and DNA. The grey arrow indicates multiple protein monomers bound to the probe.”.

In addition to this data, we have now added more EMSA results (Figure 2b) that clearly show the substrate specificity of the Prim-PoIC and its preference for binding DNA gaps rather than free DNA ends resulting from DSB formation.

3) Figure 3b and 3d. When stated that it was "observed a preference of CTP insertion", Fig. 3 b is indicated in the text. However, Fig. 3b does not address the fidelity (all 4 rNTPs are present), thus here it should be cited Fig. 3d instead. Fig.3b shows that rNTPs are better incorporated when the 5'-end is not phosphorylated, while the difference with phosphorylated template is much less pronounced. There is more unextended product with dNTPs, but also the extended products look more, as if the total amount of template was not equal between rNTPs and dNTPs experiments. If the authors want to make this as a strong point, a nucleotide titration is required in order to quantitatively assess the difference. Was the synthesis due to CMP incorporation only or was it dependent on the template sequence? The experiment shown in Fig.3d clearly indicated that with 5'phospho ends all four rNTPs and dNTPs can be incorporated opposite a template T, and it is hard to say that CMP is preferred

without proper quantification. In general, one of the major authors' conclusion is that the LigC complex preferentially carries BER using rNTPs (cited also in the discussion, p.15). However, such clear preference was not demonstrated. The reactions were not carried out with physiological levels of either rNTPs or dNTPs (which are not equimolar mixtures). In addition, no selectivity values are given, making it impossible to state what the real preference is and if under physiological conditions rNTPs may really compete with dNTPs for incorporation. A competition assay including rNTPs and dNTPs at physiologically relevant ratios should be carried out, at least in an experimental setup similar to Fig. 3d, to show the relative incorporation (the products can be distinguished by their different electrophoretic mobility). In addition, selectivity indexes should be derived for PrimPoIC rNMPs vs. dNMPs incorporation.

We fully agree. As the Reviewer astutely points out, the CTP preference observed could be due to the sequence upstream from the templating base. In light of related AEP's having demonstrable scrunching / dislocation activities, we have removed this statement and the panel, Figure 3d, from the manuscript.

In reference to rNTP vs dNTP preference, the preference for adding rNTPs over dNTPs is a well characterized feature of PrimPols related to LigD, which is consistent with their biological roles as primases/polymerases. To validate our statement further, we have now obtained and included a new set of data from nucleotide competition assays (Supplementary Figure 5a performed to quantify the nucleotide preference factor F, which is ~ 190 fold in favour of rNTP incorporation by Prim-PoIC. This compares to ~70 fold preference observed with Prim-PoID.

4) Figure 3c. The ligated products appear heterogeneous, with at least two or three molecular species migrating with decreasing mobility. If the band representing the expected product is the major one, were the other species due to extra added nucleotides? If so, what is the explanation? Does it mean that the repair with rNTPs is error-prone (low fidelity of insertion plus extra nucleotide addition?)

The DNA repair involving the activities of LigD and related enzymes is known to be a very error-prone process. It has been widely speculated it requires a further repair, directly after or during the next round of DNA replication to remove the incorporated RNA. We suggest in the discussion that PolA is the best candidate to carry out such corrections both during the double strand break repair and during short gap repairs. PrimPolC has some residual terminal transferase activity and this could contribute to the observed heterogeneity of DNA repair products that we can see.

5) Figure 6. Why deletion of ligC1 and C2 gives better tolerance to oxidative stress than deletion of ligC2? Why ligC1 (which was the main protein studied here) was not deleted alone? If ligC1 works together with PrimPolC, one should expect to observe the same increased sensitivity as the PrimPolC deleted mutant for a ligC1 deletion. The data, however, seem to indicate that PrimPolC is important for oxidative stress tolerance irrespectively from ligC1/C2 (compare the effects of ligC1/C2 deletion with the PrimPolC alone or the triple deletion).

To support our claims about the role of LigC *in vivo*, we have now performed a much more detailed and unambiguous phenotypic analysis to address this issue using more sensitive CFU survival assays on our mutants and we also generated additional mutants for comparison (Figure 6b & Supplementary Figure S8). These show a clear epistatic relationship between Prim-PolC and the LigC1, 2 consistent with our proposed model for their roles in the repair of oxidative repair. In addition, we also knocked out LigD and have also uncovered an unexpected role for this protein in excision repair that is independent of LigC. These results are discussed in the paper. The reason we chose to knock out both ligases (LigC1 & 2) together instead of singularly is because in most organisms, including most mycobacteria, there is only one LigC ligase present so we believe this is a better mimetic of a LigC KO in the other systems.

6) In the text it is not clarified whether the authors also carried out mock-purification controls for their eGFP fusion tags, in order to exclude unspecific protein binding. This should be specified.

We have previously published control datasets with eGFP only to evaluate the background that is normally observed in the eGFP facilitated purifications. Please refer to Plocinski et al. 2014 for further details.

MINOR POINTS

1) Introduction, p.2 line 9 from top: "is...forms" should be "is...that forms"; p.3, line 5 from bottom: RNA is not incorporated, rNTPs (or even better rNMPs) are. Replace RNA with rNTPs.

The errors are now corrected.

2) Results, p.5 line 4 from top: some detail about the nature of the C-ter extensions (or some relevant reference) may be of interest. p.5, line 7 from bottom: please define DSSP

The reference has now been added and the DSSP crosslinking agent is defined.

3) Figure 2a: in the Table there is a problem with the character encoding in the PDF in the row of uvrB

The error is now corrected.

4) p.6: EndoIV results may be shown as supplementary data

We have now included an additional set of data showing association of Prim-PolC, together with EndoIV on the single nucleotide gap substrates *in vivo*.

5) p.7, after "genetic interactions" Figure 2c should be cited, since it is not cited anywhere else.

Figure 2C has now been cited

6) Fig. 3b. The size markers on the right side of panel b are shifted up

The error has now been corrected

Reviewer #2 (Remarks to the Author):

This work is focused on the DNA repair apparatus called Ligase C complex. The authors showed that the LigC complex is involved in base-excision repair (BER) in mycobacteria. Moreover, their studies indicate that LigC complex interacts with BER proteins *in vivo* and together form an apparatus with a distinctive function in DNA repair (i.e. removing and mending damaged bases and abasic sites). Moreover, they show that cells lacking the LigC complex are more sensitive to oxidative DNA damage. In the current state, this work does not seem suitable for this journal. Perhaps some major revisions may help.

1. The writing style is rather difficult to follow, probably addressed to specialists rather than to a broad audience. The names of the many proteins need to be introduced in a more structured and uniform manner. For instance, they refer to "Ligase C", sometimes "LigC", sometimes to "LigC complex". It is not easy to figure out if these are alternative names for the same entity, or whether they refer to a multiprotein complex assembled on a scaffold-like subunit enzyme called LigC. The difference should be clarified right from the beginning, since this refers to key-terms of the manuscript.

We agree. The manuscript has now been significantly rewritten to address these concerns and revised to make it simpler for the non-specialist reader to follow it.

Another example of ambiguity refers to the first paragraph of the "Results" section ("Actinobacteria possess multiple.."). In the way how the text is described, it is not clear if this represents introductory background information, or the result of their own bioinformatics analyses.

We agree with this proposal and have now relocated a reduced form of this information into the introduction.

In connection to this chapter, it is not clear the relationship between the proteins depicted as arrows of different colors in Fig. 1. Do they act in the same pathway? Are they also part of the same complex/machinery? Probably the figure is supposed to be self-explanatory.

We agree. Figure 1 has now been remade from scratch to make it much clearer to the reader and put into the supplementary figures as Figure S1, given the concerns in the previous query.

2. The proteins Prim-PolC, LigC1 and LigC2 were co-purified as baits with multiple proteins involved in the BER pathway. The complexes were cross-linked with DSSP and subjected to mass-spectrometric analysis for the identification of the captured proteins. However, some important questions are not addressed. For instance, are all these listed proteins forming a single complex? Or they reflect various smaller subcomplexes that bind the same bait? To verify this aspect, one would need to check the migration of the affinity purified complexes, prior to cross-linking and mass-spec (for instance by native PAGE or gradient ultracentrifugation). Perhaps an SDS-PAGE of the affinity-purified complex should be also shown.

We have now performed pull-down type experiments using different DNA substrates (Supplementary Figure S4b), which are supportive of our statement that PrimPolC is involved primarily in repairing single nucleotide gaps, both in vitro and in vivo. From what we observe, the proteins recruited to the gap are probably chosen “on the fly”, based on the type of lesion detected. These proteins may be found on various types of lesions, in various combinations, rather than forming large pre-assembled complexes. e.g. LigD being found with Ku on DSB lesions but LigD is also found in large quantities on SSB substrates but without Ku being present.

3. Next, the authors use recombinant proteins to validate the interactions detected between bait proteins (LigC1 and Prim-PolC) and BER proteins. The interactions were evidenced by western-blotting with specific antibodies (Fig. 2b). They conclude that LigC1 acts as a key scaffolding protein required for sequential recruitment of BER proteins.

Giving the availability of the recombinant proteins, some additional, perhaps very informative experiments can be performed. For instance, they can evidence/describe the interactions by other methods, as size-exclusion chromatography), which additionally provides information about the stability of the complexes. This may also answer to questions like: (i) do all these proteins form a single machinery, that behaves as a single particle? (ii) alternatively, do these proteins associate in several distinct complexes? If yes, which proteins are associated physically? (iii) Do the proteins act transiently, one after another in a step-wise manner? If so, perhaps some competition experiments can be performed to support the description of the pathway where they act.

Additional co-IPs have been performed using His-tagged MPG and FPG with tag-free LigC, showing a strong interaction between LigC and FPG (Supplementary Figure S3b). Additional experiments were also performed to identify protein complexes isolated from the cells that bind to various different DNA substrates (Supplementary Figure S4b).

4. The authors solved the crystal structure of Prim-PolC and compared it to the previously reported Prim-PolD. The two structures are very similar, except for an additional C-terminal stretch that is present only in Prim-PolC, and highly conserved among its orthologues. This element (called Loop3) forms a layer with the canonical loops 1 and 2, on one side of the active site of the enzyme. From a technical standpoint, the structure is absolutely fine. However, the importance of loop3's location is over interpreted and unconvincing in the

absence of supporting experiments. Moreover, it is not conceptually well-linked to the flow of their story. Probably for this reason the authors do not mention the crystal structure in the abstract of the manuscript.

We have revised the text in the results sections and added a section to the discussion and abstract to highlight the importance of this motif. We have also added comparative data relevant to the role of Loop3 in binding gaps (Figure 2b, c & d). These revisions now depict a more cohesive story that better links the structural data with the biochemical findings.

One important question is whether loop3 is folded in the way it is due to the crystal contacts.

We have considered the question of crystal contacts and thank the reviewers for highlighting the point. Analysing the ASU we find that there are minimal crystal contacts where there are no more than four residues of Loop 3 that are involved in contact with another polypeptide chain. As shown in Reviewers Figure 1 (Below), the contacts do not seem responsible for the conformation adopted by Loop 3. We stand by the assertion that the contacts between Loop 1 and Loop 2 with Loop 3 constrain the edges of Loop 3 and direct the available conformations that can be adopted by Loop 3.

Even assuming that it is not involved in contacts, one should assay the activity of the enzyme in the absence of this loop. Moreover, point-directed mutagenesis will be required to support the rather detailed interpretation of the structure.

To address this query, we generated numerous constructs where Loop 3 and or helix 8 has been deleted or mutated at certain positions. Mutants generated were T1 M1-P300, T2 M1-G312, T3 M1-P319 and T4 M1-P320, and a number of point mutants. Unfortunately, despite months of effort using a variety of expression systems and refolding procedures, all of these mutations were insoluble and therefore this prevented us from performing functional assays to test the functional role of Loop 3 in DNA binding etc.

Reviewer #3 (Remarks to the Author):

Plocinski et al. present a study of prokaryotic/mycobacterial LigC, an ATP-dependent DNA repair ligase. The Doherty group and others have previously done substantial work on a related protein, LigD, and shown it to comprise a multifunctional nonhomologous end joining (NHEJ) double-strand break (DSB) repair complex active in stationary phase. In contrast to prior descriptions that LigC might function in a redundant NHEJ pathway, the current study provides evidence from a number of different experimental approaches that LigC acts as part of a base excision repair (BER) single-strand break (SSB) pathway, again most active in stationary phase. Similar to LigD, LigC incorporates rNTPs, consistent with function in a non-replicative phase when dNTPs are limiting.

The LigC, PrimPolC and related proteins are of considerable interest in important microbiology and as correlates to other species, as well as revealing of general principles in DNA repair. As such, a definitive description of a dedicated role of LigC in a unique BER pathway would be highly significant as well as novel. A strength of this study is the number of different approaches used, even including crystallography to describe a novel protein loop. The result is a collection of supportive finding that create a plausible and compelling model of LigC function. Unfortunately, in being so diverse, many specific aspects of the study are not

performed at enough depth to make them as compelling or informative as needed, so that in the end the study falls short of its claim of definitively “[establishing] a distinct DNA repair pathway required to excise and replace ... bases ... during stationary phase”.

Specific points:

1) The 2014 J. Bacteriology paper from Glickman is a clearly a central reference point for this study, as it claimed a role for LigC in NHEJ. First, it would seem relevant to reference this paper at the appropriate place in the 2nd Introduction paragraph. The Glickman study also measured LigC expression as a function of culture growth stage – how is the current study additionally informative? Most importantly, the Discussion fails to ever consider the current results in the context of the prior study, simply saying that the NHEJ connection was “not proven”, which is not helpful to readers.

This paper describing the potential role of LigC in NHEJ is now referred to in the second paragraph of the Introduction. We did not intend to diminish the findings of Glickman et al. but simply point out why we believe it is not appropriate to claim that LigC is involved in NHEJ based on these data.... LigC was never shown to operate in NHEJ in a clean LigD mutant, only in the background of a point mutant of LigD with a non-functional ligase domain. The DSB can still be processed by the working domains of LigD and the lost ligase activity supplemented in trans by LigC but this by no means proves that this is its function...

To support our claims about a specific role of LigC in excision repair processing *in vivo*, we have now performed a much more detailed and unambiguous phenotypic analysis to address this issue using more sensitive CFU survival assays on our mutants and we also generated additional mutants for a better comparison (Fig. 6b & Fig. S8). These show a clear epistatic relationship between Prim-Pol C and the LigC, consistent with our proposed model for their roles in the repair of oxidative repair. In addition, we also knocked out LigD and have uncovered an unexpected role for this protein in excision repair, that is independent of LigC. These results are discussed in the paper. Together, our data clearly establish that a major role of LigC in cells is to assisting in excision repair of oxidative lesions. We cannot 100% rule out it does not have a minor role in NHEJ but convincing evidence to support this hypothesis is still unavailable.

2) At many points, the text is very rough and in need of grammatical correction. There are too many instances to list, but a first instance can be found in the first Introduction paragraph “The best characterized AEP is PolDom forms part of a ...”

We agree. The manuscript has now been thoroughly rewritten and revised to address these concerns.

3) Figure 1 is of dubious value to the paper, and problematic in its presentation. First, is it a novel description (it seems not)? If essential to rapid understanding of the current study, it is failing that purpose as the labeling is inadequate – why wouldn't the genes be labeled with their identities as text? Are all species essential to display? Is the disposition within the genome critical for understanding this paper? Finally, the '&' symbol is meaningless without consulting the legend, which shouldn't be necessary for such a simple figure.

We agree that there are some issues with this figure but we think it helps to nicely introduce the major topic and questions addressed in the paper. With these concerns in mind, Figure 1 has now been remade from scratch, to make it much clearer to the reader, and put into the supplementary figures as Figure S1.

4) Extending the previous point, the first Results paragraph has a lot of descriptive detail that does not seem novel or essential. The point is unclear and being lost. I think it could be substantially shortened and moved to Introduction, as the purpose seems to be helping the reader understand that LigC is a structurally definable and broadly conserved protein, previously established information.

We fully agree with this point and have relocated a reduced form of this text from the Results to the Introduction, where it is more appropriate.

5) Can Fig. 2a include gene names for all entries? It is unnecessarily difficult to identify all proteins (e.g. “Deoxyribonuclease” is uninformative).

The figure now contains gene names for all entries.

6) The slot blotting method in Figure 2b is useful as a rapid way of probing many potential interactions, but is not quantitative and subject to a number of assumptions. Findings would be more compelling if confirmed individually by pull-down types of experiments. Also, while the Ku control is important, it might be useful to include LigD, given previous finding that LigC can cooperate during NHEJ.

The co-IPs have now been performed with His-tagged MPG and FPG and tag-free LigC, showing a strong interaction between LigC and FPG (Supplementary Figure S3b). Additional experiments were also performed to show protein complexes isolated from the cells, binding to various DNA substrates (Supplementary Figure S4b).

7) In Figure 3a, the arrows and bands are never defined. I gather the presumption is that the black arrow denotes a specific complex at the nick/gap, and the grey arrow denotes a non-specific complex; has this been validated?

The arrows are now defined.

8) Figure 3b is difficult to understand. First, it is lacking a critical no-enzyme control. Second, the 0/+1/etc. labels do not appear to line up with the correct bands, and are very distant from the leftmost lanes in any case. I was ultimately able to convince myself that the gel is understandable if the enzyme adds 1 nt to a nick, but it took me too long to get there. Finally, the depiction of the overhang/ov substrate is not intuitive; I’m still not certain what it is.

Figure 3 has now been modified to address these issues.

9) Figure S2b lacks any lanes to establish the position of the expected product from the DSB ligation substrates. Among other things, this could include incubation with LigD. Also, these lanes never seem to be mentioned anywhere – while they aren’t definitive proof of a lack of DSB ligation ability of LigC, or of a preference for nicks (given that any ligase can ligate a nick), it seems relevant to mention.

To address this query regarding a potential role of LigC in DSB repair, we now include additional biochemical evidence to show Prim-PolC activity is not primarily connected to DSB processing. Significantly in this regard, and in contrast to the NHEJ Prim-PolD, we show that it lacks significant affinity towards DNA overhangs (Figure 2b) and is unable to mediate break synapsis (Figure 2c) or perform MMEJ-dependent synthesis (Figure 2d). These data, combined with our other results support its role in excision, and not DSB, repair process in mycobacteria.

10) Figure 3d is never mentioned and without discussion is difficult to understand. I think perhaps it was intended to be omitted and is related to a comment at an earlier point in the text? I agree with omitting it, it doesn't strongly support the most important points of the paper.

Figure 3d was indeed an oversight and has now been removed from the manuscript.

11) In Figure 4b the authors have tried to be too clever in their labeling and made it harder, not easier, to understand what ultimately is a conceptually simple experiment, that ligated products can be seen when all required lesion processing activities are added. To the extent that I believe I ultimately understand the experiment, it cannot be considered revealing beyond what is known about their individual enzymatic activities. In the absence of more careful studies of combinations, order of addition, amounts, kinetics etc. we learn little additional about the enzymes, and of the functional interactive complex, which is being claimed. Even the simplest controls like lanes omitting LigC are absent. And, what if LigD were added instead of LigC, i.e. how specific is this to LigC?

The major overall goal of this study was to define the cellular pathway to which the LigC complex belongs. Whilst we fully agree that a more detailed and thorough biochemical / kinetic dissection of this enzymatic repair pathway(s) is required, this will take a number of years to complete and goes beyond the remit of the current study. We are presently undertaking this programme of work and look forward to reporting our findings in future publications over the next few years.

12) The structural studies of LigC are the most compelling part of the manuscript and very interesting. I think the very long paragraph describing the loops should be broken up to make the different important insights more digestible. Figure 5a seems to have lost a "Loop 3" label that would be helpful to add back. Also, the authors never actually define Loop 3 in the text – they just start using it at one point. Labeling of the 5' phosphate and binding pocket will also help orient readers, as would labeling the inferred DNA:loop 3 steric conflict. Ultimately, though, while the interesting structure of LigC loop 3 almost certainly does alter its catalytic behavior, insufficient data are provided to develop a binding model for a nick/gap substrate – things are currently largely conjectural.

We have now fixed this issue in Figure 5a and 5d. The definition of Loop 3 is now explicitly defined in the text and, we have changed the flow of the text to highlight the relationship between Loop 3 and the innate ability of the enzyme to bind to gapped substrates in preference to DSBs.

We have also added extra panels (Figures 2b, c & d) where we demonstrate this biochemically in comparison to Prim-PolD (PolDom). We hope these additions remove the conjectural notion that the reviewer refers to.

13) If Figure 6b, it is never clearly stated why TBH and CHP were used, as opposed to other agents (H₂O₂?) -is there something important about organic oxidizing agents? Also, I find the text description of the results very confusing and difficult to match up to the figure. To me, the text implies construction of strains that are not shown (e.g. a combined PolD2 Prim-PolC mutant). It is also unclear and not addressed why ligC2 alone seems to have a larger effect than ligC1,ligC2 combined. Also, why is PrimPolC more sensitive than ligC1,ligC2? The rest of the paper might give the impression that LigC is the defining protein of the pathway. Finally, to support the claims of the paper it would be important for comparisons of sensitivity to be made at different growth stages.

CHP induces the highest induction of PolA of all tested chemicals based on TBDB database. This was one of the reasons we picked it as an oxidizing agent as it is an established BER inducer. We initially tested a number of oxidizing agents, including peroxyntrite and H₂O₂. These substances however, tend to degrade instantly when added to the cultures, whereas in the presence of organic peroxides we were able to get more pronounced phenotypes after elongated incubation times. We choose these to best visualize the phenotypes of the mutants. We have now included survival results using CFU methodology, which is probably more informative for the reader. The phenotypes with H₂O₂ are now shown in as a supplemental figure.

14) Also regarding the sensitivity to oxidative damaging agents, while there are certainly single-strand lesions there may also be double strand lesions, so it can be difficult to be certain which lesions are responsible for LigC mutant sensitivity. Further study of genetic interactions would help interpretation, including with LigD and well as with other SSB repair components, where different outcomes of synergy and epistasis might be expected. In other words, can it be bolstered that the *in vivo* phenotypes are truly due to SSB, not DSB repair?

We fully agree with these concerns. To address these and support our claims about the role of LigC complex *in vivo*, we performed a much more detailed and unambiguous phenotypic analysis to address this issue using more sensitive CFU survival assays on our mutants and we also generated additional mutants for comparison (Figure 6b & Supplementary Figure S8). These show a clear epistatic relationship between Prim-Pol C and the LigC1,2 consistent with our proposed model for their roles in the repair of oxidative repair. In addition, we also knocked out LigD and have also uncovered an unexpected role for this protein in excision repair, that is independent of LigC. These results are discussed in the paper.

15) The discussion of why it makes biological sense to have an RNA-incorporating BER pathway in stationary phase is clearly relevant and important to the Discussion but quite long; I think this could be tightened up considerably. It seems to have been engaged at the exclusion of consideration of any of the structural biology, or of a clear comparison of current to previous studies.

We fully agree and have rewritten the current discussion to make it more succinct and added new sections discussing key points arising from both the the structural and phenotype studies.

Reviewers Figure 1. Display of symmetry related molecules in contact with Loop 3. **(a)** A ribbon representation of Prim-PolC molecules that are within contacting distance with residues of the C-terminal extension from the central molecule. Solvent accessible surfaces cover the regions involved with electrostatic surface colourings for the areas of contact with Loop 3. The colour scheme for the central molecule is as described in main text Figure 5a. **(b)** A zoomed in view of **(a)** Loop 3 with the close contacts from two other symmetry-related molecules.

Reviewers' Comments:

Reviewer #1:

Remarks to the Author:

In this revised version of the manuscript, the authors provide additional data to corroborate their observation of a role of PrimPolC in BER. A few minor things should still be clarified.

1) Answer to my point 1. Supp. Figure 3b. Those are pull-downs not co-IPs (as far as I understood no Abs have been used). The data are fine, it would be just easier for the reader to understand the experiment, if in the Figure or Figure Legends it was specified that the pull-downs were performed with MPG-his bound to beads as bait and untagged LigC as prey.

2) Answer to my Point 2 about former fig.3 (now fig. 2). The authors maintain their opinion that a quantification of the EMSAs is not necessary, in view of the clear difference between the unphosphorylated/phosphorylated gapped substrates. While I appreciate that the analysis is qualitative and that there is a clear difference, I was under the impression that the substrate specificity of PrimPolC was a central point of the paper. The novel data in fig. 2b are also interesting in terms of the supposed activity of PrimPolC in NHEJ, that the authors object. Again, giving a quantitative measure of the difference, even if just based on the quantification of the relative amounts of shifted substrate, at comparable amounts of proteins, will be useful. The data are already there, it would be just a matter of making densitometric scanning. Even better would be to derive a curve as a function of PrimPolC and PrimPolD concentrations, to estimate the apparent binding affinities. In case, the authors may choose other methods to measure the affinity of PrimPolC to different substrates. My point here is that a quantitative assessment of the DNA substrate selectivity of PrimPolC is a crucial point of the paper.

3) It is not clear whether in the new Fig. 2b the shifted complex is a monomer-DNA (as could be assumed by the use of a black arrow) or a multimer-DNA. In any case, the pattern seems different from fig. 2a (only one shifted complex of different EM with respect to either of the two shown in fig. 2a). Please clarify better in the text or figure legend how the detected complexes compare between fig. 2a and fig. 2b.

4) In fig. 3a it appears that the labelling of the order of products is wrong (+2/+1 should not be +1/+2 ?)

Reviewer #2:

Remarks to the Author:

For the revised manuscript "DNA Ligase C and Prim-PolC participate in base excision repair in mycobacteria", the authors provide clear and satisfying responses to our initial concerns and comments, and have adjusted their manuscript accordingly. The manuscript is now appropriate for publication. A few minor editorial comments are included below.

- Introduction, row 50: "..DNA ligase domains that, together with Ku, coordinate..". I would mention what "Ku" is, like "the Ku repair factor.." or at least "the Ku protein..".
- Figure 1c: one should label each of the three clusters of proteins (represented by three ovals with distinct colors).
- Discussion, row 358: "Prim-PolC preferentially binds to, and fills..". I would reformulate this sentence, perhaps even split it in two, for the sake of clarity.

Reviewer #3:

Remarks to the Author:

Plocinski et al. submit a revised version of their manuscript proposing a base excision repair (BER) role for mycobacterial LigC, which is extensively rewritten and includes new data. The text has addressed the majority of prior issues and is much improved (although there are still a number of small grammatical errors). As mentioned below, the new data do help, but overall I still find the study to fall short of its claim of definitively establishing that LigC defines a novel BER pathway. There are many intriguing findings but the inability to connect all of the dots continues to make this manuscript more suited to a specialized journal.

1) The major elements of new or revised data panel include:

a) a pull-down experiment (in Supplemental) between LigC and Fpg. The design of this experiment is a bit difficult to follow, and the methods are never described– the reader is left to infer what was done. The LigC-Fpg interaction does seem robust, but the Mpg does not, despite imprecise word choices like “weaker” that imply something different than what I see in the figure. The pull-down pattern is thus not predicted by the far western, so isn’t strongly corroborating and a more careful and extensive interaction analysis is still warranted. The far western remains the central panel in the main paper, which I don’t think will be convincing to most readers.

b) Further analysis of PrimPolC using DSB substrates. Overall, this is a nice addition to the paper, and sets up one of the most substantiated portions of the discussion, that this paper has documented important biochemical and structural differences between PrimPol C and D, an important contribution.

c) A supplemental panel summarizing results (in a supplemental table) identifying protein that bound to different type of DNA substrates in a lysate pull-down. This is a potentially very interesting addition, but nearly impossible to evaluate as presented. The figure is just a final interpretation, but the data in the Excel file are very difficult to figure out without detailed careful attention, and even so, the quantitative value of the method is not clear, i.e. how strongly are different proteins present? It might be very interesting if it could be claimed that LigD bound SSBs without Ku, in contrast to DSBs, but I cannot judge this in the current presentation.

d) A supplemental panel (worthy of the main paper?) showing the rNTP preference. No concerns here.

e) Redo of the oxidation sensitivity using CFU assays, including addition of comparison to LigD. First, the authors claim an epistatic relationship, but this requires construction and testing of a double mutant. The text claims a mutant of PrimPol-C, LigC1, LigC2 on page 14, but it is never discussed further, and is absent from all figure panels. Regarding LigD, while this is an important addition, it might be considered to work against the claims of the paper. While SSB vs. DSB lesion levels are addressed in Discussion, I still think the biological readout might lack the desired specificity for SSB lesions. On the other hand, if LigD is also an even more potent contributor to excision repair, this undermines the apparent importance of LigC and continues to suggest it is a “back up” more than a pathway uniquely evolved to support BER.

2) Issues that I don’t think were well attended to from my prior critique:

a) Figure 3a still lacks a critical no-enzyme lane, in the absence of which it is very difficult to understand what the enzyme has actually done. The quantitative comparison of rNTP to dNTP in panel is still difficult, although other evidence above makes this case well. The previously missing band labels have been added, although are apparently incorrect (the order is 0,2,1,3,4)

b) The labeling scheme in Figure 4 is still unnecessarily difficult. Also, the figure is less valuable than the authors seem to think. Comparison of outcomes to LigD (or even T4 ligase?) would be one example of a valuable addition that would help understand the specificity of interactions. As it stands, the figure simply demonstrates that a polymerase fill gaps that can be ligated by a ligase,

which reveal little about a true biological pathway.

c) Regarding the crystal work. It remains very strong and interesting, but the fundamental conjecture is still there. Specifically, while is an attractive model that Loop3 is responsible for an SSB-specific function of PrimPolC, there is no evidence provided to support that claim. It is unfortunate that mutants have not been tractable yet, but this doesn't change the lack of proof of a structure-function relationship with Loop 3.

Reply to reviewers' comments:

We would like to thank all the reviewers for reading the revised manuscript, particularly given it is the holiday season, and for their constructive comments on our revisions. We have provided responses to their additional queries below.

Reviewer #1 (Remarks to the Author):

In this revised version of the manuscript, the authors provide additional data to corroborate their observation of a role of PrimPolC in BER. A few minor things should still be clarified.

1) Answer to my point 1. Supp. Figure 3b. Those are pull-downs not co-IPs (as far as I understood no Abs have been used). The data are fine, it would be just easier for the reader to understand the experiment, if in the Figure or Figure Legends it was specified that the pull-downs were performed with MPG-his bound to beads as bait and untagged LigC as prey.

The text has now been modified accordingly to the reviewer's comment. The revised version of the supplementary legend is shown below.

Supplementary Figure 3. Operonic association of bifunctional glycosylase FPG with Prim-PolC and LigC homologues in mycobacteria.

(a) For chosen mycobacteria, genomic regions encoding FPG co-transcribed with either both, or a single Prim-PolC and LigC homologues, in the vicinity of base excision repair genes *polA* and *uvrB*, are presented. (b) Pull-down assay confirmation of strong protein-protein interactions between the recombinant, tag-free LigC (the prey protein) and FPG-6xHis (the bait protein) from *M. smegmatis*, with respective controls.

2) Answer to my Point 2 about former fig.3 (now fig. 2). The authors maintain their opinion that a quantification of the EMSAs is not necessary, in view of the clear difference between the unphosphorylated/phosphorylated gapped substrates. While I appreciate that the analysis is qualitative and that there is a clear difference, I was under the impression that the substrate specificity of PrimPolC was a central point of the paper. The novel data in fig. 2b are also interesting in terms of the supposed activity of PrimPolC in NHEJ, that the authors object. Again, giving a quantitative measure of the difference, even if just based on the quantification of the relative amounts of shifted substrate, at comparable amounts of proteins, will be useful. The data are already there, it would be just a matter of making densitometric scanning. Even better would be to derive a curve as a function of PrimPolC and PrimPolD concentrations, to estimate the apparent binding affinities. In case, the authors may choose other methods to measure the affinity of PrimPolC to different substrates. My point here is that a quantitative assessment of the DNA substrate selectivity of PrimPolC is a crucial point of the paper.

The quantitative analysis of Prim-PolC binding to the gapped substrate, originally requested by the reviewer 1, has now been included in the figure.

3) It is not clear whether in the new Fig. 2b the shifted complex is a monomer-DNA (as could be assumed by the use of a black arrow) or a multimer-DNA. In any case, the pattern seems different from fig. 2 a (only one shifted complex of different EM with respect to either of the

twos shown in fig. 2a). Please clarify better in the text or figure legend how the detected complexes compare between fig. 2a and fig. 2b.

We now make a clear distinction in the concentration ranges used for EMSA's in Figures 2a & b both in the figure itself and in the figure legend. The supershift band (denoted by the grey arrow) is only seen at concentrations of Prim-PolC above 5 μ M, whereas the concentration range used for Figure 2b reaches a maximum of 100nM which is 50 times lower than the concentration at which the supershift is seen.

4) In fig. 3a it appears that the labelling of the order of products is wrong (+2/+1 should not be +1/+2 ?)

Thanks for spotting this error. The labelling has now been corrected.

Reviewer #2 (Remarks to the Author):

For the revised manuscript "DNA Ligase C and Prim-PolC participate in base excision repair in mycobacteria", the authors provide clear and satisfying responses to our initial concerns and comments, and have adjusted their manuscript accordingly. The manuscript is now appropriate for publication. A few minor editorial comments are included below.

- Introduction, row 50: "..DNA ligase domains that, together with Ku, coordinate..". I would mention what "Ku" is, like "the Ku repair factor.." or at least "the Ku protein..".

The text is now modified accordingly.

- Figure 1c: one should label each of the three clusters of proteins (represented by three ovals with distinct colors).

The clusters are now labelled.

- Discussion, row 358: "Prim-PolC preferentially binds to, and fills..". I would reformulate this sentence, perhaps even split it in two, for the sake of clarity.

The sentence has been reformulated as requested.

Reviewer #3 (Remarks to the Author):

Plocinski et al. submit a revised version of their manuscript proposing a base excision repair (BER) role for mycobacterial LigC, which is extensively rewritten and includes new data. The text has addressed the majority of prior issues and is much improved (although there are still a number of small grammatical errors). As mentioned below, the new data do help, but overall I still find the study to fall short of its claim of definitively establishing that LigC defines a novel BER pathway. There are many intriguing findings but the inability to connect all of the dots continues to make this manuscript more suited to a specialized journal.

1) The major elements of new or revised data panel include:

a) a pull-down experiment (in Supplemental) between LigC and Fpg. The design of this experiment is a bit difficult to follow, and the methods are never described— the reader is left to infer what was done. The LigC-Fpg interaction does seem robust, but the Mpg does not, despite imprecise word choices like “weaker” that imply something different than what I see in the figure. The pull-down pattern is thus not predicted by the far western, so isn’t strongly corroborating and a more careful and extensive interaction analysis is still warranted. The far western remains the central panel in the main paper, which I don’t think will be convincing to most readers.

A description of this method is now included. The discrepancies between the strength of signal in far western and pull-down may be caused by a number of factors i.e. end-blocking while MPG is bound to the beads. FPG is by far the strongest interactor of LigC, based on its co-transcription with LigC-like and Prim-PolC-like proteins in other mycobacteria and direct, strong interaction *in vitro* we are showing. Also, it should be noted that FPG, LigC and Prim-PolC proteins are present in very low abundance in the cell, which makes studying their interactions much more difficult *in vivo*, due to the detection limits for such un-abundant complexes.

b) Further analysis of PrimPolC using DSB substrates. Overall, this is a nice addition to the paper, and sets up one of the most substantiated portions of the discussion, that this paper has documented important biochemical and structural differences between PrimPol C and D, an important contribution.

We thank the reviewer for this positive comment.

c) A supplemental panel summarizing results (in a supplemental table) identifying protein that bound to different type of DNA substrates in a lysate pull-down. This is a potentially very interesting addition, but nearly impossible to evaluate as presented. The figure is just a final interpretation, but the data in the Excel file are very difficult to figure out without detailed careful attention, and even so, the quantitative value of the method is not clear, i.e. how strongly are different proteins present? It might be very interesting if it could be claimed that LigD bound SSBs without Ku, in contrast to DSBs, but I cannot judge this in the current presentation.

The supplementary excel file (Supp. Table ST3) presenting the label free quantification of the proteomic data has now been modified for improved clarity. The results are presented in the standard format provided by the quantification software, here – MaxQuant, and the intensity values correspond to the relative abundance of each protein in the sample. The Ku protein is completely absent from the SSB substrate, whereas LigD is present at high amounts, together with Prim-PolC. This finding is even more potent when one considers the naturally low abundance of Prim-PolC in mycobacterial cells.

d) A supplemental panel (worthy of the main paper?) showing the rNTP preference. No concerns here.

No action required.

e) Redo of the oxidation sensitivity using CFU assays, including addition of comparison to LigD. First, the authors claim an epistatic relationship, but this requires construction and testing of a double mutant.

The mutant lacking Prim-PolC, LigC1, LigC2 was constructed and included in the original version of the manuscript, when we used spot assay rather than CFU methodology to assess sensitivity of our mutants to oxidizing agents. Based on this previous result, we claimed the epistatic effect. However, we found including all the created strains in the figure to be confusing and not very helpful for the reader, crowding and diluting the final message, and for clarity, have removed them from the recent version of the manuscript. We have now also removed the claim of the epistatic effect of Prim-PolC and the ligases as we no longer show data to support this claim. However, we have presented data for a double mutant (Δ LigD, *prim-polC*) that shows these pathways are non-epistatic in their survival profiles.

The text claims a mutant of PrimPol-C,LigC1,LigC2 on page 14, but it is never discussed further, and is absent from all figure panels.

This was included in error (see above) so we have removed this mutant from the text as it was not used in the revision.

Regarding LigD, while this is an important addition, it might be considered to work against the claims of the paper. While SSB vs. DSB lesion levels are addressed in Discussion, I still think the biological readout might lack the desired specificity for SSB lesions. On the other hand, if LigD is also an even more potent contributor to excision repair, this undermines the apparent importance of LigC and continues to suggest it is a “back up” more than a pathway uniquely evolved to support BER.

The effect we observed on survival in stains lacking LigDs vs. Prim-PolC during oxidative repair is not epistatic so LigC is certainly not acting as a back-up pathway for LigD or vice versa. Removal of both proteins causes a significant sensitization of the tested strains to oxidizing agents. This observation is an important and highly significant finding as previous studies failed to show significant phenotypes for Prim-PolC mutants. The effect is apparent even with low amounts of H₂O₂, known not to induce significant amounts of double strand breaks. Rather than weakening our findings, it enhances the story and shows that the picture is more complex than originally propose. There is growing evidence that LigD is also very much implicated with oxidative repair and our experiments support this. However, as stated in the discussion, future detailed studies are required to define the specific lesions that each pathway deals with and it will be exciting to see if LigD is specific to SSBs and LigC to base damage etc.

2) Issues that I don't think were well attended to from my prior critique:

a) Figure 3a still lacks a critical no-enzyme lane, in the absence of which it is very difficult to understand what the enzyme has actually done. The quantitative comparison of rNTP to dNTP in panel is still difficult, although other evidence above makes this case well. The previously missing band labels have been added, although are apparently incorrect (the order is 0,2,1,3,4)

Figure 3a has now been amended to show the no-enzyme lane and the band labels are now in the correct order. The qualitative comparison of rNTP to dNTP preference we feel is obvious in this panel and the reviewer rightly acknowledges the other evidence provided making the case.

b) The labeling scheme in Figure 4 is still unnecessarily difficult.

We have simplified the labelling scheme in 4B and 4C by removing the symbols and replacing them with abbreviations for the enzymes added to the reactions to make it clearer for the reader.

Also, the figure is less valuable than the authors seem to think. Comparison of outcomes to LigD (or even T4 ligase?) would be one example of a valuable addition that would help understand the specificity of interactions. As it stands, the figure simply demonstrates that a polymerase fills gaps that can be ligated by a ligase, which reveals little about a true biological pathway.

Showing the ability of purified DNA repair proteins to co-operatively repair their natural DNA substrate *in vitro* is critical to support the model proposed. However, simply adding proteins to reconstitute such processes *in vitro* is by no means a trivial task, as their comments implicitly implies, and certainly not guaranteed to work, especially given the “power” of the nucleases we are adding, that can completely destroy the substrates. For instance, when the main BER-pathway polymerase - PolA was used in the same way to this assay, the DNA substrates were completely destroyed, instead of being repaired. The pathway we are showing here critically requires minimal DNA end-resection to maintain the overall integrity of the DNA. The FPG-EndoIV-LigC-Prim-PolC complex ensures minimal gap processing and insertion of very short patches of rNTPs, to avoid rNTPs mediated DNA instabilities. Clearly, further work is now required to add more detail to this mechanism and better define the enzymatic steps and interactions involved at each stage of this repair process but this goes beyond the scope of this study.

c) Regarding the crystal work. It remains very strong and interesting, but the fundamental conjecture is still there. Specifically, while it is an attractive model that Loop3 is responsible for an SSB-specific function of PrimPolC, there is no evidence provided to support that claim. It is unfortunate that mutants have not been tractable yet, but this doesn't change the lack of proof of a structure-function relationship with Loop 3.

Whilst we agree that the role of Loop 3 remains to be established, which we explicitly acknowledge in both the results and the discussion, a better understanding of its function will minimally require the elucidation of a co-crystal complex with DNA. While this work is in progress, it will be some time before it is completed so we believe, given this is a relatively minor part of the current manuscript, that it is important to publish the initial story about LigC's role in excision repair and then focus on more mechanistic aspects in follow up publications.

Reviewers' Comments:

Reviewer #1:

Remarks to the Author:

In this second revision the authors have added the requested clarifications/missing data. I have no further remarks.

Reviewer #3:

Remarks to the Author:

Płociński et al. have attended to remaining presentation issues in the text and figures. In rebuttal, they have also made some reasonable arguments in response to my previous comments on the significance and interpretation of presented data. Because this revision does not contain new data, my overall impression remains unchanged regarding the incomplete degree to which the data fully and compellingly support the claim of a novel base excision repair pathway. However, stationary phase repair is an interesting and important subject in microbiology and genetics, and a number of interesting observations are made with strong data regarding especially PrimPolC. As such, the data are worthy of publication, to let readers decide and to stimulate further work in the area.